# Neurovascular coupling and $CO_2$ interrogate distinct vascular regulations

Marine Tournissac [1,2] ✉, Emmanuelle Chaigneau [1,5], Sonia Pfister [3,5], Ali-Kemal Aydin [1], Yannick Goulam Houssen[1], Philip O'Herron[4], Jessica Filosa [4], Mayeul Collot [3], Anne Joutel [2] & Serge Charpak [1] ✉

Neurovascular coupling (NVC), which mediates rapid increases in cerebral blood flow in response to neuronal activation, is commonly used to map brain activation or dysfunction. Here we tested the reemerging hypothesis that $CO_2$ generated by neuronal metabolism contributes to NVC. We combined functional ultrasound and two-photon imaging in the mouse barrel cortex to specifically examine the onsets of local changes in vessel diameter, blood flow dynamics, vascular/perivascular/intracellular pH, and intracellular calcium signals along the vascular arbor in response to a short and strong $CO_2$ challenge (10 s, 20%) and whisker stimulation. We report that the brief hypercapnia reversibly acidifies all cells of the arteriole wall and the periarteriolar space 3–4 s prior to the arteriole dilation. During this prolonged lag period, NVC triggered by whisker stimulation is not affected by the acidification of the entire neurovascular unit. As it also persists under condition of continuous inflow of $CO_2$, we conclude that $CO_2$ is not involved in NVC.

Human brain functional imaging uses functional hyperemia, a rapid and local increase in blood flow, to map brain regions activated by sensory stimulation and behavioral tasks, as well as assess brain connectivity. The signaling pathways by which neurovascular coupling (NVC) generates functional hyperemia involve the cooperation of numerous cell types, and it has long been accepted that these pathways are triggered by neurotransmitter feedforward mechanisms rather than from local metabolic feedback due to acute oxygen and glucose consumption, or $CO_2$ production[1,2]. Several arguments were successively raised and put aside in support of the metabolic feedback hypothesis. Initial measurements of partial pressure of oxygen ($pO_2$) with Clark electrodes, performed in the brain parenchyma of anesthetized animals with acute craniotomies[3–5], reported that sensory stimulation generated local $pO_2$ dips, the timing of which was compatible with a triggering role in NVC[6]. This hypothesis was, however, recently questioned, as the pO2 dip disappeared in chronically prepared awake mice, i.e., more akin to physiological conditions[7]. The triggering role of oxygen in NVC is therefore unlikely, even though one

study reported that local application of $O_2$ scavengers could generate a $pO_2$ dip and an increase in blood flow[8]. Concerning glucose, hyperglycemia was recently shown to result in decreased NVC in the zebrafish optic tectum[9], whereas other experiments have reported that NVC is unaffected by hypoglycemia[10,11]. The role of $CO_2$ production as a trigger of NVC was initially discarded because synaptic activation in vivo generates an alkalinization of the extracellular fluid[12–14], which should constrict, rather than dilate, smooth muscle cells (SMCs)[15]. On the other hand, in the mouse cortex, the extracellular alkalinization can be reversed to a rapid acidosis upon blockade of the sodium bicarbonate co-transporter 1 (NBCe1) in astrocytes[13], suggesting that the dynamics of $CO_2$ production, normally buffered by astrocytes, is fast enough to participate in NVC. Because some studies also suggested that $CO_2$ per se, independent of pH, has a direct effect on SMCs relaxation during cerebrovascular reactivity to $CO_2$ ($CVR_{CO2}$), and that $CO_2$, in addition to passive diffusion through cell membranes can diffuse through selective gas channels (for review see ref. 16), the role of $CO_2$ production in NVC has reemerged. In particular, Hosford et al.

[1]Sorbonne Université, Inserm U968, Vision Institute, Paris, France. [2]Université Paris Cité, Institute of Psychiatry and Neuroscience of Paris (IPNP), INSERM U1266, 75014 Paris, France. [3]Chemistry of Photoresponsive Systems, Laboratoire de Chémo-Biologie Synthétique et Thérapeutique (CBST) UMR 7199, CNRS, Université de Strasbourg, Strasbourg F-67400 Illkirch, France. [4]Department of Physiology, Augusta University, Augusta, GA, USA. [5]These authors contributed equally: Emmanuelle Chaigneau, Sonia Pfister. ✉e-mail: marine.tournissac@inserm.fr; serge.charpak@inserm.fr

recently used a protocol they described as occlusion or saturation protocol to investigate the issue in rodents[17], in which, upon breathing $CO_2$ (10%) over a 10-min period, cerebrovascular responses to electrical stimulation of the paw were abolished, even when resting blood flow was maintained at a constant rate with indomethacin and caffeine. They proposed that exogenous $CO_2$ diffusing from the bloodstream was saturating $CO_2$ dependent processes, i.e., masking the effect of $CO_2$ produced by neuronal metabolism. Endogenous $CO_2$ would thus mediate a feedback mechanism involving NBCe1, leading to $CO_2$ removal through functional hyperemia. As both NVC and $CVR_{CO2}$ are widely used in clinics, this controversial hypothesis requires further support and characterization of the possible sites of action of $CO_2$ produced by neurons along the vascular arbor. In this work, we synthetize a vascular pH sensor and use the barrel cortex neurovascular model, in chronic sedated mice, to investigate the interactions between NVC and $CO_2$. We first characterize a $CO_2$ stimulus, brief and strong (subsequently referred to as brief$CO_2$), that triggers vascular responses within only a couple of seconds. We then precisely measure the onsets of GCaMP6/8 fluorescence changes in cells of the neurovascular unit, and pH and blood flow responses along the vascular arbor following brief$CO_2$. The analysis of CVR to brief$CO_2$ and NVC response onsets and interactions, at both the microscopic and mesoscopic level, indicate that $CO_2$ does not play a role in NVC.

## Results

### brief$CO_2$ stimulation causes early and delayed responses

To investigate the potential role of $CO_2$ in NVC, we applied a systematic approach based on timing measurements of physiological responses, which enabled us to uncover compartmentalization of vascular responses during NVC[18]. In contrast to many previous works that studied $CVR_{CO2}$ using long lasting $CO_2$ stimulations, on the order of several minutes, to reach a steady state (for reviews on $CVR_{CO2}$ see ref. [19–21]), we tested the effect of brief$CO_2$, a 10 s inhalation of 20% $CO_2$ to generate sharp, reversible and reproducible vascular and cellular responses. All experiments were done after a 7–10 day recovery period from surgery and under sedation with dexmedetomidine, as we found that brief$CO_2$, even at 5% $CO_2$, is too stressful to train awake mice. Under sedation, responses to brief$CO_2$ could be reproduced up to 20 times per experiment (at 5 min inter-trial intervals), and over months without any observable adverse effects.

Figure 1 illustrates that brief$CO_2$ triggered an early increase of breathing frequency about 2 s after the beginning of $CO_2$ application (mean onset of $1.8 \pm 1$ s, $n = 19$ animals, 30 experiments), whereas it required about 3-4 additional seconds to dilate both pial and penetrating arterioles (mean onset for all arterioles, $5.8 \pm 1.4$ s, $n = 23$ animals, 29 experiments; see also Supplementary Movie 3). Note that pial arteriole diameter was directly measured using two-photon microscopy line-scan acquisitions, whereas for penetrating arterioles the diameter was calculated from the lumen area measured with movie acquisitions (see Methods). As baseline diameter fluctuates at 0.1 Hz due to vasomotion[22,23], vessel diameter changes were expressed in z score to improve the precision of dilation onsets, and to minimize the contribution of trials and vessels with the largest 0.1 Hz diameter fluctuations (see our previous work[24] and Methods). Dilation onsets were determined using an approach based on fits (onset is defined as the time to reach 10% of the peak of the fit; see Methods).

As dilation can be associated with an increase or decrease in blood velocity[18], we performed simultaneous measurements of the pial arteriole diameter and blood velocity with broken line-scans (Fig. 2a). Figure 2b illustrates that surprisingly, brief$CO_2$ was followed by an early decrease in velocity (average onset of $2.9 \pm 0.6$ s, $n = 7$ vessels, 6 mice). As a result, blood flow (calculated as $\pi R^{2*}$ velocity) increased over resting flow with a large delay of $7.5$ s $\pm 2.4$ s (time to reach 10% over the baseline; $n = 7$ vessels, 6 mice). Dilation and velocity changes

amounted to about 10% (Fig. 2b inset; see also Supplementary Fig. 1). Note that in 4 additional penetrating arterioles that had a small segment parallel to the image plane (i.e., in which velocity could be measured), brief$CO_2$ was also followed by an early decrease in blood velocity (average onset of $3.4 \pm 0.9$ s; $n = 4$ vessels, 3 mice), demonstrating that pial and penetrating arterioles respond similarly to brief$CO_2$. The uncoupling between blood flow and velocity responses was markedly different from what occurs during neurovascular coupling, where sensory stimulation triggers nearly simultaneous increases of both velocity and vessel diameter in pial arterioles (see ref. [24]). Thus, depending on the type of vascular signal reaching the pial arteriole, backpropagating during NVC or propagating downstream during brief$CO_2$, the regulation of blood functional parameters strongly differs.

The decrease in velocity during the 3–4 s period where the arteriole diameter remains constant implies that it results from an upstream event, e.g., occurring at the level of the middle cerebral artery, the circle of Willis, or the internal carotid. To support this hypothesis, we imaged the dynamics of cerebral blood volume (CBV) in depth using functional ultrasound imaging (fUS), an approach now well established in anesthetized and awake mice[25–27]. Figure 2c–e shows that brief$CO_2$ resulted in an early drop of 10% of CBV in the internal carotid (mean onset of $2.0 \pm 1.3$ s; $n = 6$ experiments, 5 mice), which preceded the CBV increase in the cortex by 3 s (mean onset of $4.7 \pm 1.5$ s; $n = 5$ experiments, 5 mice, time to reach 10% over the baseline). Alignment of fUS CBV responses in the carotid and velocity responses in pial arterioles indicate that the CBV signal in the carotid underlies the early velocity drop (with constant diameter) in cortical arterioles (Fig. 2e inset). The carotid signal itself resulted from a transient and modest ( ~ 1 mmHg) drop of arterial pressure, which occurred rapidly during brief$CO_2$ (Supplementary Fig. 2c). To conclude, brief$CO_2$ resulted in early (velocity) and delayed (diameter) arteriole responses, with the early response being compatible with the time required for blood, acidified and transporting $CO_2$, to reach the carotid and brain arterioles[28].

### pH/$CO_2$ underlies early GCaMP6/8 fluorescence changes in cells composing the arteriole NVU

To investigate which cell type contributed to the arteriolar dilation induced by brief$CO_2$, we measured the dynamics of GCaMP6 and GCaMP8 fluorescence in each cells of the arteriole neurovascular unit (NVU), i.e., in endothelial cells, SMCs, astrocyte end-feet and perivascular neurons in different strains of transgenic animals (Fig. 3a). Surprisingly, brief$CO_2$ decreased fluorescence in all cells of the NVU. In transgenic mice expressing GCaMP6 in SMCs, brief$CO_2$ resulted in a decrease in fluorescence 2–3 s before dilation (Fig. 3b, onset from $CO_2$ application: $3.3 \pm 0.7$ s; $n = 6$ vessels, 5 mice), contrasting to the rapid decrease that occurs during neurovascular coupling (Fig. 3a inset; see also ref. [18]). In astrocytes and endothelial cells, this decrease in onset was variable but preceded arteriole dilation (endothelial cells: $3.3 \pm 1.5$ s, $n = 8$ vessels, 4 mice; astrocytes: $2.1 \pm 0.6$ s, $n = 5$ vessels, 4 mice). The dendritic neuropil onset ($3.9 \pm 1.3$ s; $n = 7$ vessels, 5 mice) was not significantly different from the dilation onset. As GCaMP6/8 fluorescence is known to decrease with acidosis[29,30] and since fluorescence decreased in all NVU cells, we raised the hypothesis that brief$CO_2$ caused a decrease of intracellular pH rather than of intracellular calcium. We synthetized and injected intravenously a modified form of the fluorescent pH sensor H-Ruby[31] attached to a dextran (70 kDa) (Fig. 3c). The H-Ruby dextran conjugate displayed a 53-fold fluorescence enhancement at 580 nm (from pH 10 – 4.5) with a measured pKa of $7.57 \pm 0.06$. It did not leak from blood and, when co-injected intravenously with AlexaFluor488 (AF488, not sensitive to pH) also attached to a 70 kDa dextran, allowed for the quantification of vascular pH changes from fluorescence ratio measurements (see Methods and Supplementary Fig. 3). Figure 3d shows that brief$CO_2$

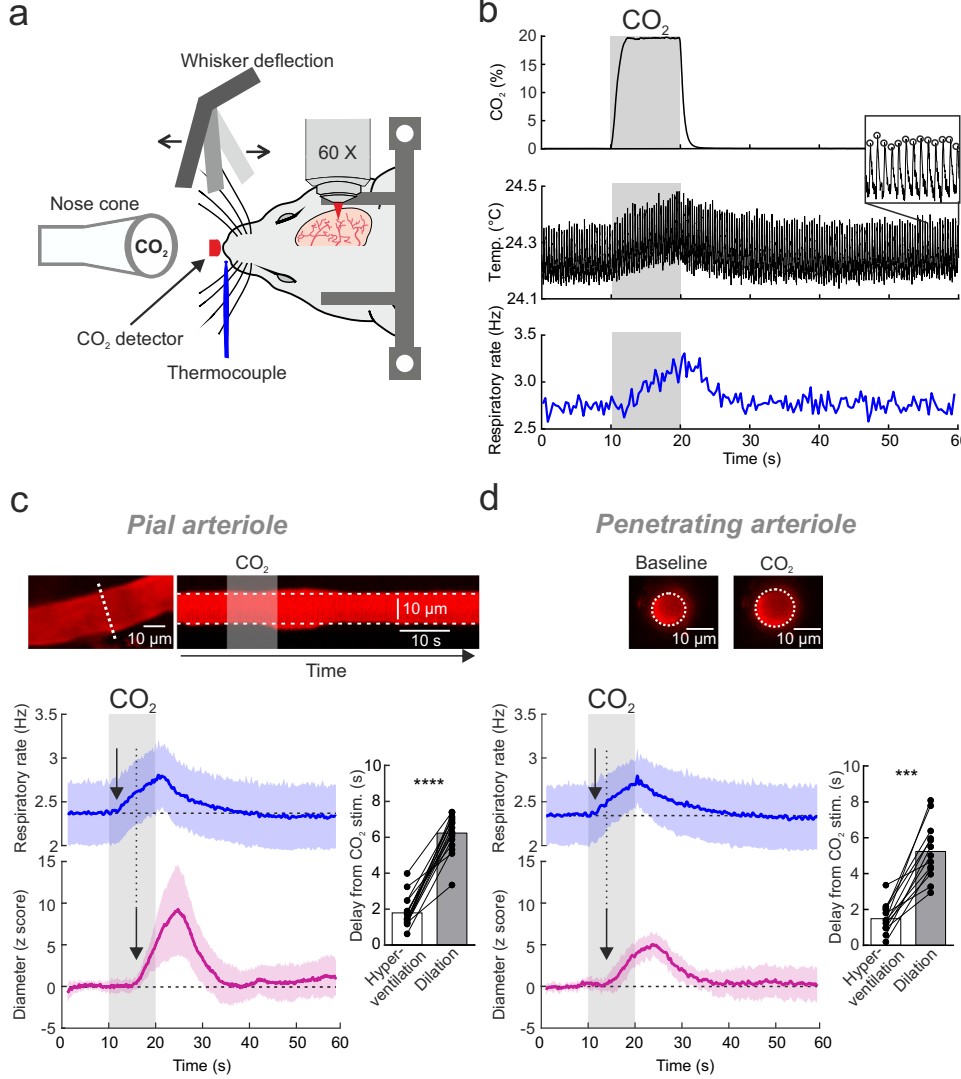

**Fig. 1 | briefCO₂ stimulation generates early and delayed responses. a** Head-fixed sedated mice, implanted with a chronic cranial window over the barrel cortex, were exposed to a brief and strong hypercapnic stimulation (briefCO₂: 20%, 10 s) or to whole pad whisker stimulation (5 Hz, 5 s). Respiration was monitored with a thermocouple placed in front of the nostril and blood flow imaged with two-photon imaging. **b** Top, CO₂ concentration measured at the nostril reached a plateau within 2 s. Middle and inset, detection of temperature peaks enabled the respiratory rate to be computed (bottom, single trial), which increased after a delay of about 2 s. **c** Top left, a pial arteriole labeled with Texas Red injected i.v. The diameter was measured with line-scan acquisitions drawn perpendicular to the vessel (white dotted line). Top right, Representative example of a pial arteriole dilation upon briefCO₂. Bottom left, Average responses to briefCO₂. Diameter changes are expressed in z score to minimize the contribution of spontaneous 0.1 Hz fluctuations. Dilation (pink trace) lagged the hyperventilation (blue trace) by ~ 4 s (see the arrows, paired data, $n = 16$ arterioles, 12 mice, 2–5 stimulations per vessel). Bottom right, quantification of the onsets (two-sided Wilcoxon rank sum test, ****$p < 0.0001$). **d** Top, the diameter of penetrating arterioles was calculated from lumen area. Bottom left and right—as for pial arterioles, hyperventilation (blue trace) precedes dilation (pink trace) by ~ 4 s (paired data, $n = 13$ arterioles, 11 mice, 2–5 stimulations per vessel, two-sided Wilcoxon rank sum test, ***$p = 0.0002$). The arrows indicate the onset time to reach 10% of the fit of the curves. Data are represented as mean ± SD (shadings).

resulted in an increase in H-Ruby fluorescence (i.e., a pH decrease) in pial arterioles after a delay of $2.0 ± 0.7$ s ($n = 5$ vessels, 4 mice). Note that as opposed to the CO₂ delivery at the mouse nostril which plateaued within 2 s (Fig. 1b), the pH response did not saturate (Fig. 3d, lower left). The shape of the signal can be taken as a temporal marker of the inflow of acidified blood transporting CO₂ and reaching the arterioles. It thus explains the early signals triggered by briefCO₂: the hyperventilation, the blood velocity decrease and NVU cell decreases of GCaMP6/8 fluorescence. Based on the properties of H-Ruby, we estimated that vascular pH decreased by about 0.15 pH on average (see Methods), a very significant change but brief enough to not threaten the animal's vital prognosis. Overall, the results show that briefCO₂ strongly acidifies cells of the entire arteriolar NVU with a dynamic fast enough to investigate the role of pH/CO₂ in NVC.

## CO₂ does not interact with NVC measured at the arteriolar and capillary level

NVC involves several signaling pathways that may differ according to the site of neural excitation, i.e. at the level of arterioles or capillaries[32]. The onset of NVC occurs within ~ 1–2 s in rodents, with a time to peak between ~ 1–2 s for short stimulations and is modulated by anesthesia[24,33]. There is a consensus that in addition to the activation of astrocytes and pericytes, synaptic activation increases local potassium concentration, which generates a backpropagating signal along the vascular arbor, dilating large capillaries (also named the transitional segment[34]), and penetrating and pial arterioles[18,35]. In the somatosensory cortex, long CO₂ stimulation (10%, 10 min) was shown to result in markedly decreased NVC, an effect attributed to the saturation of CO₂-dependent processes (i.e. to an "occlusion" of the response) triggered

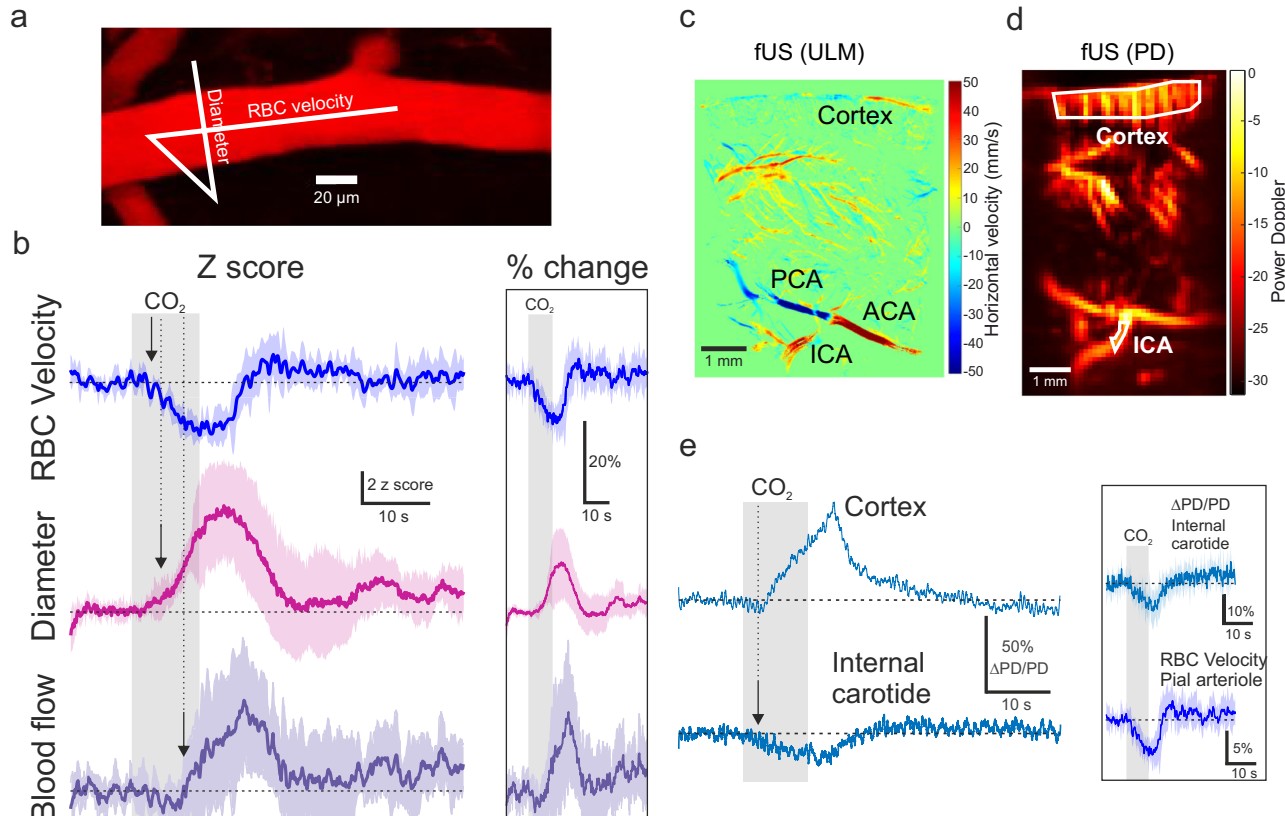

**Fig. 2 | Uncoupling of arteriole diameter, blood velocity and blood flow during briefCO₂.** **a** Top, red blood cell (RBC) velocity and diameter were measured simultaneously with a broken line-scan drawn along and across the pial arteriole (white line). **b** Upon briefCO₂, velocity (blue trace) decreases before the delayed arteriole dilation (pink trace). Calculated blood flow (velocity * $\pi$.radius²) (purple trace) shows a transient decrease followed by an increase due to dilation. Right inset shows data presented in percent change from baseline ($n = 7$ vessels, 6 mice). **c** A sagittal section of a mouse brain was recorded with ultrasound localization microscopy (ULM) upon microbubbles injection (i.v.). This approach allowed to position CBV measurements in the internal carotid artery (ICA). Note that it also

reported resting blood velocity, the color map representing the horizontal component of the RBC speed. **d** Power Doppler image of the same sagittal section acquired with standard functional ultrasound imaging (fUS). A high-pass filter selecting the CBV flowing with an axial velocity >8 mm/s was applied. **e** Left, briefCO₂ generated a delayed increase in CBV in the cortex preceded by a small early drop, concomitant with an early CBV decrease in the ICA (average of 5 stimulations). Inset, comparison of the dynamics of CBV responses in the ICA (top, $n = 6$ experiments, 3–5 stimulations, 5 mice) and RBC velocity responses in pial arterioles (bottom, same trace as in panel b). The arrows indicate the onset time to reach 10% of the fit of the curves. Data are represented as mean ± SD (shadings).

by neuronal activation and resulting in NVC[17]. The timing of briefCO₂ makes it possible to test the hypothesis that exogenous CO₂ alters NVC as CO₂ diffusion and acidification of the periarteriolar column precedes arteriole dilation by several seconds. NVC occurring within 0.5–1 s of neuronal activation (Fig. 3a inset) following whisker stimulation (5 Hz, 5 s) was triggered either simultaneously with CO₂ inhalation (before the arteriole acidification, i.e., control conditions) or 2 s after CO₂ inhalation (during NVU acidification) (Fig. 4a, b). Figure 4c shows that neuronal responses (perivascular bulk calcium; Thy1-GCaMP6s mice) were not affected during CO₂ diffusion and acidification of the periarteriolar column. Note that the drop of GCaMP6 fluorescence following neuronal activation resulted from CO₂-induced acidosis and not from a delayed inhibition, as briefCO₂ alone triggered the same drop of fluorescence and neuronal stimulation alone was not followed by a delayed fluorescence drop. In addition, the superposition of arteriole dilations in both conditions show that the first phase of dilation (the first 5–6 s leading to the dilation peak) was similarly not affected by CO₂ (Fig. 4d). To analyze the extent to which NVC and CO₂ vascular responses were additive or not, we compared the calculated summation of these responses measured separately (Fig. 4e) with the experimental responses to both stimuli (i.e. experimental summation), with whisker stimulation occurring either simultaneously with briefCO₂ (Fig. 4f) or with a 2 s delay (Fig. 4g).

Figure 4f–h shows that the calculated summation of separate responses and the experimental summation with the two stimulation paradigms are similar when considering the first 5-6 s of NVC (up to the response peak). This means that there is perfect additivity and no occlusion of the two responses during this period. Thus, the early period of CO₂ diffusion and acidification of the NVU does not influence the initiation of NVC, although it may influence the second phase of NVC (see Supplementary Fig. 4 and Discussion).

We then tested whether NVC is altered at the capillary and the arteriolar levels during a long CO₂ stimulation, as in Hosford et al[17]. by evaluating RBC velocity responses in the capillary bed (vessels located ≥ 5th branching order) and blood flow responses in pial arteries during 14 min of CO₂ stimulation (10%, Fig. 4i). In addition to increased respiration rate and decrease of pH (-0.1 unit, see Supplementary Fig. 5), resting RBC velocity and blood flow slowly increased during the first minute and stabilized after 4–5 min (+39% ±19 and +62% ±25, respectively) (Fig. 4j, o; see also Supplementary Movie 2). Even under this condition of higher blood flow due to long hypercapnia, whisker stimulation generated similar RBC velocity and neuronal responses at the level of the capillary bed (Fig. 4k–n), as well as similar dilation and blood flow responses at the level of the arteries (Fig. 4p-s). Our approach suggests that, at the level of single arterioles and capillaries, CVR_{CO₂} does not interact with NVC.

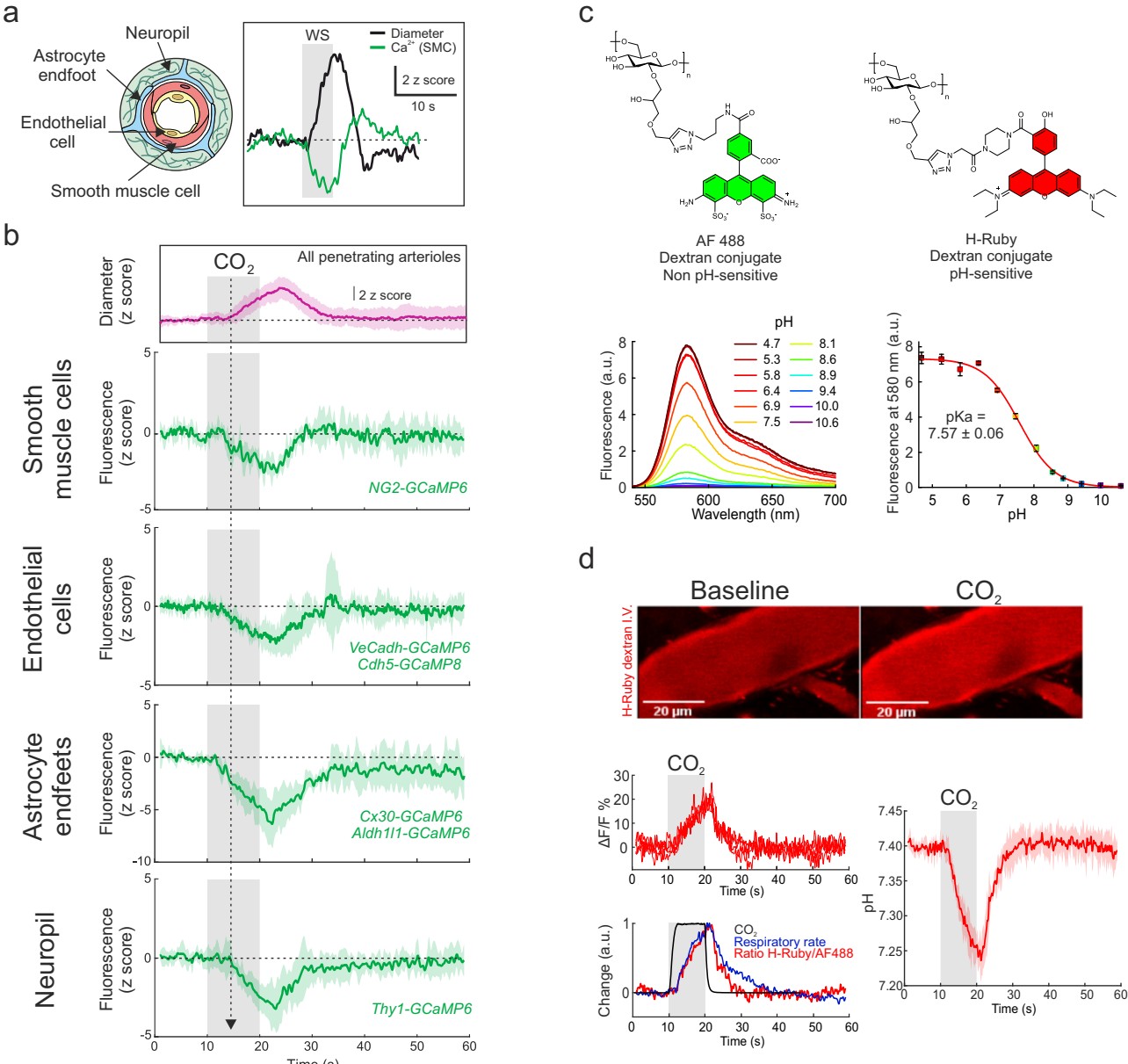

**Fig. 3 | briefCO₂ acidifies the whole neurovascular unit of penetrating arterioles. a** Left - Schematic representation of the different cells composing the neurovascular unit. Right – The inset illustrates the observation that during neurovascular coupling (whisker stimulation, WS), GCaMP6 fluorescence from smooth muscle cells (SMCs) decreased over a few hundred milliseconds (green trace) before dilation (black trace) ($n = 4$ vessels, 3 mice). **b** Upon briefCO₂ stimulation, dilation of the penetrating arteriole (pink trace) was preceded by a decrease in GCaMP6 or GCaMP8 fluorescence (green traces) in all cells of the neurovascular unit (note that fluorescence of each cell type was recorded individually in one of the six transgenic lines expressing either GCaMP6 or GCaMP8), suggesting that it may result from modulation of GCaMP fluorescence due to pH ($n = 6$ vessels, 5 mice for smooth muscle cells; 8 vessels, 4 mice for endothelial cells; 5 vessels, 3 mice for astrocytes; 7 vessels, 5 mice for the neuropil). The dotted arrow indicates the onset

time to reach 10% of the fit of the dilation curve. Data are represented as mean ± SD (shadings). **c** Top, H-Ruby (pH-sensitive) and AF488 (pH insensitive) fluorescent molecules functionalized to Dextran 70 kDa. Bottom left, fluorescence emission spectra of H-Ruby as a function of pH. Bottom right, H-Ruby fluorescence intensity curve as a function of pH ($n = 3$ measurements). Arbitrary units (a.u.). Data are represented as mean ± SD. **d** Top, H-Ruby fluorescence at rest (left) and during briefCO₂ (right). Bottom (top left), 5 consecutive responses to briefCO₂ showing a reproducible increase in fluorescence in plasma. Bottom left, superposition of normalized traces of respiratory rates (blue trace), H-Ruby/AF488 fluorescence ratio (red trace) and inhaled CO₂ concentration (black trace). Arbitrary units (a.u.). Right, briefCO₂ caused a reversible pH decrease (0.15 pH units) with an onset of about 2 s ($n = 5$ vessels, 4 mice). Data are represented as mean ± SD (shadings).

## CO₂ does not affect NVC measured at the regional level

In humans, NVC and CVR$_{CO2}$ are mostly evaluated with BOLD fMRI, i.e., with mesoscopic resolution. We thus used fUS to investigate the extent to which CO₂ interacts with CBV responses to whisker stimulation in the whole barrel cortex. We first evaluated the contribution of the arteriolar versus capillary compartments by comparing responses of slow (0.5–1.5 mm/s) and fast (>3.5 mm/s) flowing CBV (i.e., the CBV

fraction flowing with a low or high axial velocity[36]) to either whisker stimulation or CO₂ inhalation. Using co-registered TPLSM and fUS imaging in a single voxel, we have previously shown that slow flowing CBV is an excellent reporter of capillary responses during NVC[37]. Figure 5a, b shows that in contrast to whisker stimulation, briefCO₂ led to an increase in fast, but not slow, CBV in the barrel cortex. This is in contrast to measurements at the capillary level with two-photon

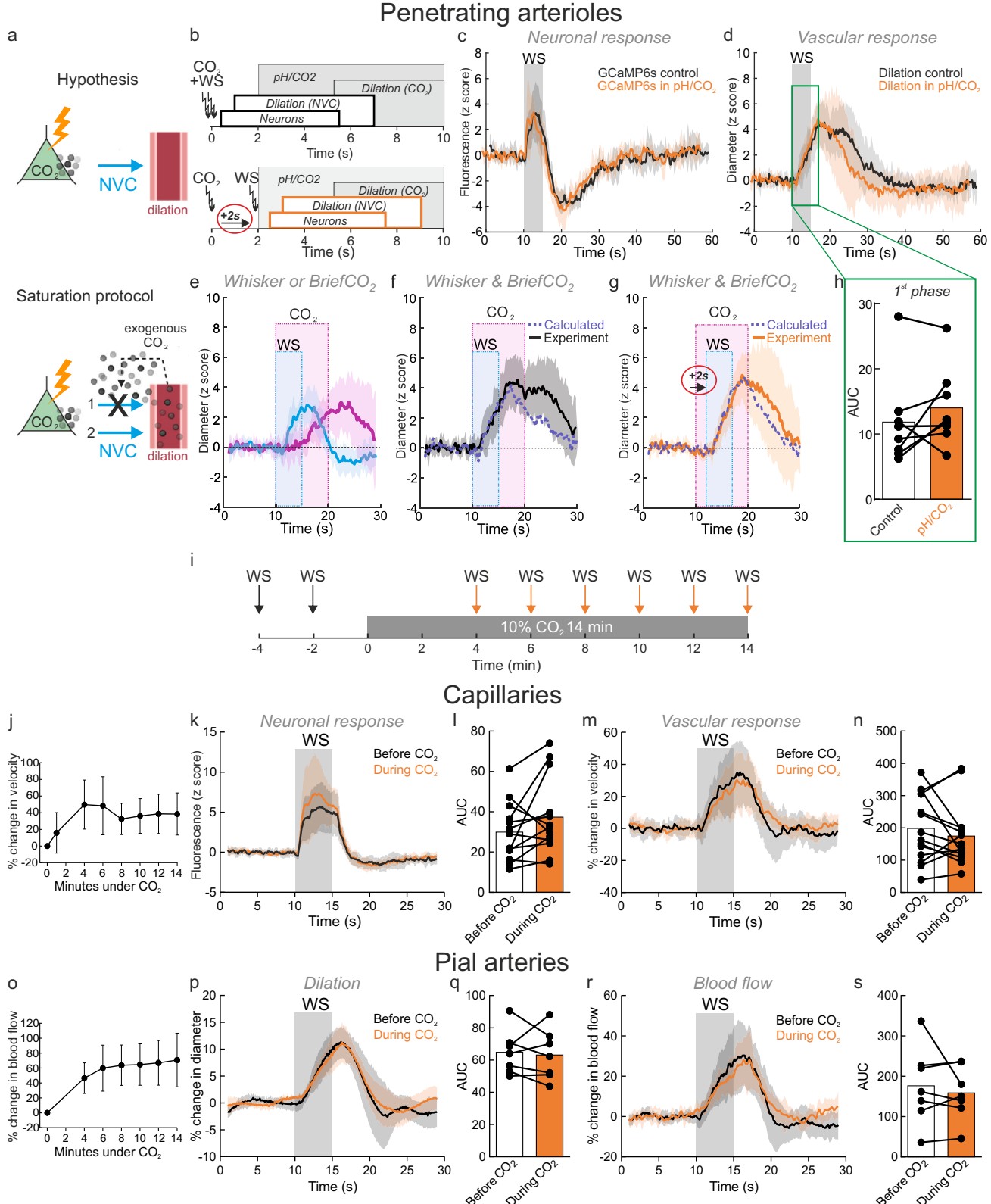

microscopy (Fig. 5c), which revealed small and delayed RBC velocity responses. This suggests that slow vascular responses to $CO_2$ are too small to be detected with fUS. Note that the dynamic of RBC velocity responses mimicked blood flow (arteriole level, Fig. 2b) and fast CBV (Fig. 5d) responses. Finally, we tested the effect of 14 min of $CO_2$ (10%) stimulation; Fig. 5d–f shows that functional hyperemia persisted during the prolonged $CO_2$ stimulation, i.e. on top of increased resting

blood flow due to $CO_2$, and that neither fast nor slow CBV responses were affected by $CO_2$. This demonstrates that NVC and $CVR_{CO2}$ do not interrogate the same cerebrovascular reflexes.

## Discussion

Here, we address the role of $CO_2$ produced by neuronal activation in generating NVC in layer 2/3 of the barrel cortex. All of our experiments

**Fig. 4 | CO₂ does not interact with neurovascular coupling measured at the single vessel level. a** Schematic of the hypothesis: Top, $CO_2$ produced by neurons initiates NVC. Bottom, exogenous $CO_2$ (during briefCO₂ or continuous $CO_2$ stimulation) diffuses from vessels to the parenchyma and either (1) saturates $CO_2$-dependent mechanisms, blocking NVC or (2) does not affect NVC. **b** Paired stimulation paradigm: top, briefCO₂ and whisker stimulation (WS) are triggered at the same time; neuronal activation and dilation due to NVC occurred before $CO_2$ reached the arteriole NVU. Bottom, whiskers were stimulated 2 s after the onset of briefCO₂. As a result, NVC occurred during $CO_2$ diffusion and acidification of the arteriole wall and its surrounding neuronal parenchyma. **c** Neuronal responses (increase in $Ca^{2+}$) remained constant during briefCO₂. Note that the secondary decrease in GCaMP6 fluorescence was due to acidosis ($n = 5$ vessels, 4 mice). **d** BriefCO₂ does not affect the initiation of dilation due to NVC ($n = 9$ vessels, 4 Thy1Gc6 mice in which $Ca^{2+}$ was simultaneously measured and 2 C57BL/6 mice). **e** Vascular responses to WS (blue trace) or briefCO₂ (pink trace) measured separately ($n = 8$ vessels, 5 mice). **f** Calculated summation of the vascular responses in e (dotted purple line) and experimental data (black solid line) of the responses to whisker and briefCO2 stimulations applied at the same time. **g** Calculated

summation (dotted purple line) and experimental data (orange solid line) when whisker stimulation is delayed by 2 s from the onset of briefCO₂. **h** AUC for area under the curve during the initiation of NVC (green box) calculated between 10 and 17 s for control and 12–19 s for pH/CO2 condition considering the 2 s delay. Paired data, Wilcoxon sum rank test, ns, $p = 0.25$, $n = 8$ vessels, 5 mice. The first 7 s of NVC response are similar in both conditions (**f, g**) showing a perfect additivity of the two responses. **i**, Continuous $CO_2$ stimulation paradigm: WS was applied before or during prolonged hypercapnia (10% $CO_2$ for 14 min). **j** Upon continuous $CO_2$, resting capillary RBC velocity progressively increased, reaching a plateau at 4 min ($n = 13$ vessels, 6 mice). **k, l** Neuronal responses were similar before and during $CO_2$ exposure ($n = 13$ vessels, 6 mice). **m, n** Functional hyperemia was not affected by long $CO_2$ exposure ($n = 14$ vessels, 7 mice). **o** Resting blood flow increased in pial arteries upon continuous $CO_2$ ($n = 7$ vessels, 3 mice). **p–s** Dilation and blood flow increase in response to whisker stimulation was similar before and during prolonged hypercapnia ($n = 7$ vessels, 3 mice). **l, n, q, s** Area under the curve (AUC) between 10 and 20 s (paired data, two-sided Wilcoxon sum rank test, ns). In all graphics, data are represented as mean ± SD (shadings).

were performed in chronic rather than acute preparations, as they preserve better brain metabolism[7]. We introduce briefCO₂, a specific short and strong $CO_2$ stimulation that triggers sharp vascular and cellular responses in brain vessels and cells, and allows repetitive applications. We used an analysis based on response onsets, which previously revealed the compartmentalization of NVC[18,24]. Our results reveal that in response to briefCO₂, vascular and cellular responses occur with two main response times, an early response within ~2 s of $CO_2$ inhalation, and a delayed response within ~6 s of $CO_2$ inhalation.

Early responses included the RBC velocity drop in brain arterioles, the increase in respiration rate, the decrease in pH measured in arterioles and the decrease in GCaMP fluorescence in all cells of the NVU. fUS measurements in the internal carotid show that the arteriole velocity drop may result from an early decrease of fast CBV in the carotid, which occurs concomitantly with an early decrease of mean arteriole pressure. We did not further investigate the mechanisms underlying the early CBV drop as it did not involve the brain per se. The onset of hyperventilation was rapid. Considering the speed of blood flow and the vascular distances for blood, loaded with $CO_2$, to reach either brain stem respiratory centers or the barrel cortex, we can assume that with the temporal resolution of our measurements, the pH drop occurs with similar onsets in the two brain regions. Similarly, we cannot distinguish which of peripheral or central chemoreceptors are first involved in the response[38–41]. The important point, rather, is that the vascular pH indicates that blood transporting $CO_2$ reaches brain arterioles within 2 s, a delay compatible with the time required for blood, acidified in the lung, to reach the cortex[28]. Acidification of NVU cells was fast. Our measurements are indirect as they are based on decreased fluorescence of GCaMP6/8 in NVU cells. Because this decrease occurs in all cells and it is known that pH changes, as large as those caused by briefCO₂ (0.15 of pH unit), can lead to a decrease in fluorescence of GFP or $Ca^{2+}$ protein sensors such as GCaMP or R-GECO[29,42], thus we believe that they report intracellular acidosis rather than true calcium changes, although some calcium changes may be masked by acidosis, in particular in SMCs. In these cells, the drop of calcium due to NVC starts < 400 ms before arteriole dilation[18]. This brief delay is thus 7 times smaller than the delay (~3 s) occurring between the fluorescence drop and dilation due to briefCO₂. It indicates that if some calcium decrease truly occurs upon $CO_2$, it is masked and involves a different signaling pathway than that triggered during NVC. Note that it is also possible that $CO_2$ affects the frequency of some calcium hot spots in astrocytes, endothelial cells or SMCs, as our approach was not developed for their observation. A previous in vivo study reported that 10% $CO_2$ stimulation (30 s) generated large and slow increases of OGB1 fluorescence in cortical astrocytes[43]. We recorded only a decrease in reporter signal, a difference that can be

partially ascribed to the fact that our $CO_2$ stimulation was brief and our mice were sedated and not deeply anesthetized with α−chloralose. Overall, the pH-sensitivity of fluorescent sensors can lead to misinterpretations but can also be used to monitor intracellular acidosis.

Delayed responses to briefCO₂ include significant increases in arteriole diameter and flow, but only minor increases in blood velocity in capillaries. This contrasts with the strong velocity response in capillaries to whisker stimulation. These findings are supported by fUS measurements, which show a strong global increase of CBV flowing at high velocity, but no detectable response for CBV flowing at low velocity in response to briefCO₂. During NVC, both slow and fast CBVs increase. The small increase of blood velocity in the capillary bed during briefCO₂ is intriguing. It may result from blood redistribution throughout the entire capillary bed, as opposed to the more confined increase occurring during NVC. Although our CBV measurements do not support the hypothesis, it could also involve a blood redirection indepth, as $CO_2$ dilation was suggested to be more important in arterioles from deep layers than from layer 2/3[44]. It is thus important to consider that NVC and $CVR_{CO_2}$ explore different vascular beds, in addition to the fact that $CVR_{CO_2}$ also shows a regional specificity[45].

In rodents, NVC onset occurs within ~1–2 s with an initial response peaking at about ~1–2 s (for brief stimulations) and, depending on the stimulation strength, a delayed response that involves astrocytes[46,47]. In our work, we have principally investigated the role of $CO_2$ on the initial component of NVC, the goal being to determine whether $CO_2$ is a necessary mediator to trigger NVC. We find that NVC remains robust under the two conditions of $CO_2$ stimulation. (1) briefCO₂, a strong stimulation during which $CO_2$ diffusion through the blood brain barrier must be massive, rapid, and limited in time. Our measurement of the intracellular acidosis onset in a given cell type indicates the maximum time necessary for $CO_2$ to reach that cell. In other words, acidosis may vary in amplitude and dynamics according to the expression of transporters and channels participating in cell-specific pH regulation. The fact that the decrease in fluorescence observed seemed faster in astrocytes than in endothelial cells or SMCs, despite astrocytes being further from the arteriole wall, indicates that $CO_2$ had already passed the first two cell types and that astrocytes have specific machinery to rapidly buffer brain acidification[13,48]. Therefore, if $CO_2$ per se has an effect on these three cells types, which participate in triggering local NVC, the initial phase of NVC should have been affected. In the case of perivascular neurons, it may be more complex; although probable, there is no proof that some $CO_2$ already diffused within the periarteriole dendritic neuropil when they responded to whisker stimulation, given that acidosis was not yet detected. It is thus not surprising that the neuronal response remained constant. However, NVC clearly occurred during acidosis and $CO_2$ diffusion: NVC and

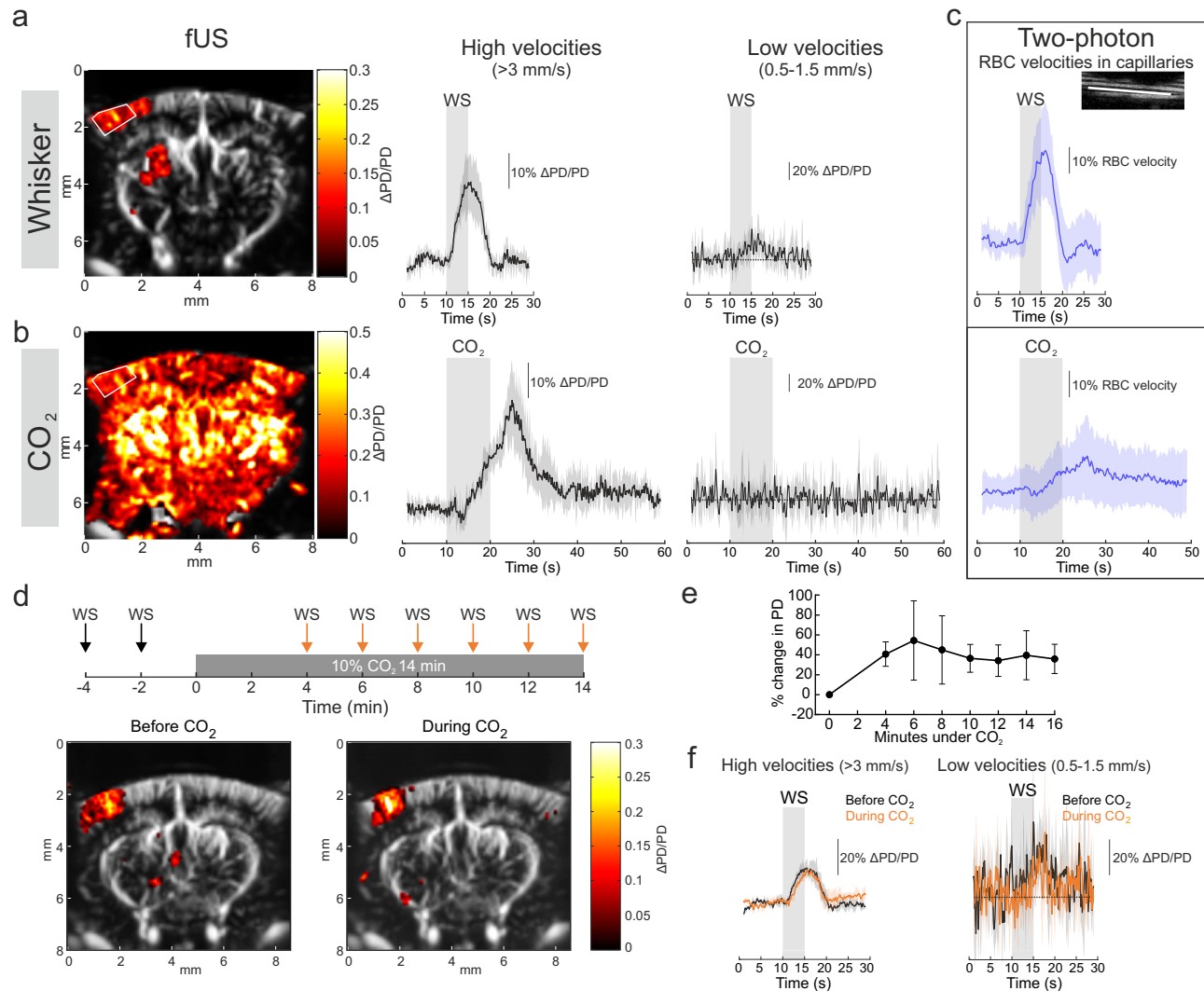

**Fig. 5 | CO₂ does not interact with neurovascular coupling measured at the regional level. a** Left, activated voxels (ΔPD/PD) in response to whisker stimulation (WS) superimposed on the Power Doppler image of a coronal section of the mouse brain (average of 4 stimulations). Right, ΔPD/PD responses to WS in the barrel cortex (ROI framed by the white lines). Axial velocities were filtered to separate CBV flowing at high (>3 mm/s) and low (>0.5–1.5 mm/s) speed, corresponding to large and small (capillaries) vessels respectively (n = 6 experiments, 5 mice). Note the small responses at low velocity. **b** Left, activation map in response to briefCO₂ showing a widespread activation throughout the brain (average of 4 stimulations, same brain section as above for WS). Right, ΔPD/PD responses for high and low velocities (same ROI as for WS) showing no detectable responses to briefCO₂ in small vessels (n = 5 experiments, 5 mice). **c** Responses to briefCO₂ were however detectable in capillaries with two-photon microscopy (n = 44 vessels, 25 mice), although much smaller than during WS (n = 23 vessels, 15 mice). **d** Top, schematic of the occlusion paradigm. Bottom–activated voxels (ΔPD/PD) superimposed on the Power Doppler map showing similar response to WS applied before (left) or during (right) prolonged CO₂ stimulation (10%, 14 min). **e** Resting CBV increased upon continuous CO₂ application as represented by the percent change in resting Power Doppler (n = 4 experiments, 3 mice). **f** Increased CBV to WS for high (left) and low (right) axial velocity is similar in control or during prolonged hypercapnia (n = 4 experiments, 3 mice). Data are represented as mean ± SD (shadings).

CO₂ responses showed full additivity during the first 6 s of NVC. A delayed second phase (or hump) appears during paired stimulation experiments. The shape of the hump depends on the timing of the summation. However, the difference between experimental and calculated summation of both NVC and CO₂ responses suggests a possible modulation of this late phase by acidosis (Supplementary Fig. 4c). Although the investigation of this possible modulation is beyond the scope of our study, many mechanisms could be hypothesized, such as the degree of actin/myosin filament overlap in SMCs at the onset of second dilation or an effect of CO₂ on the mechanisms underlying the undershoot that terminates the NVC response. (2) Prolonged 10% CO₂ stimulation, which should discard the objection that potential CO₂-dependent processes were not altered or saturated by briefCO₂. Surprisingly, NVC was not affected even after 14 min, although resting CBV and RBC velocity increased in both the arteriolar and capillary

compartments. This implies that dilated arterioles could further dilate under neural activation. This is in line with some previous studies based on 5% an 12% concentrations of inhaled CO₂[49,50], but does not support the recent work of Hosford et al. using prolonged 10% CO₂[17]. A major difference between the two studies is that our whisker stimulation is a mild sensory stimulation while paw electrical stimulation likely stimulates pain fibers in addition to sensory fibers, and it is known that hypercapnia elevates pain threshold and depresses nociception[51]. Another difference may reside in our chronic sedated mouse preparation versus the α-chloralose anesthetized ventilated rat model. In our preparation, animals were free to hyperventilate to compensate acidosis and intravascular pH dropped by 0.1 pH unit while parenchymal pH decreased by 0.2 unit in Hosford's experimental conditions during prolonged CO₂ stimulation. This could also account for some of the differences. Still, even if our acidosis is lighter, it is

important and NVC is perfectly preserved in the different vascular compartments. CVR to $CO_2$ and the signaling pathways involved have been thoroughly investigated in rodents and humans[19–21,52–54]. Here, we restricted our study to the analysis of timing. briefCO2 stresses that $CO_2$ rapidly affects both peripheral and central circulation. The timing of events observed revealed that relaxation of SMCs lagged by ~3-4 s after the drop in GCaMP6 fluorescence (i.e., cell acidosis), in contrast to what occurs during NVC, where relaxation occurs ~400 ms after a true decrease in calcium[18]. The signaling mechanisms linking acidosis to relaxation are thus distinct, slow and beyond the scope of our study. Overall, our data suggest that hypercapnia and thus probably endogenous $CO_2$ produced by neurons do not affect NVC, which principally depends on transmitter release and a rise in extracellular potassium[1,2,35]. In addition, it shows that NVC and CVR to $CO_2$ should be considered as two stimuli exploring distinct cerebrovascular reflexes with different dynamics, NVC being rapid and CVR to $CO_2$ requiring a couple of seconds to occur and probing different vascular compartments.

## Methods

### Animals
All animal care and experimentations were performed in accordance with the INSERM Animal Care and Use Committee guidelines and approved by the ethical committee (Charles Darwin, comité national de réflexion éthique sur l'expérimentation animale n°5) (protocol number #27135 2020091012114621) and animal care and use committee of Augusta University. Mice were fed ad libitum and housed in a 12-h light-dark cycle at 22 °C and 50% humidity. A total of 66 adult mice, both males and females, from 3 to 10 months of age were included. The following mouse lines were used and bred in our animal facility: C57/BL6J, NG2-CreER™, Ai95(RCL-GCaMP6f), Thy1-GCaMP6s (GP4.3), Cdh5-GCaMP8 (#033342, Jackson Laboratory), VeCadh-CreERT2, Ai95(RCL-GCaMP6f), Cx30-CreERT2 were crossed with Ai95-GCaMP6f mice to generate Cx30-CreERT2; Ai95-GCaMP6f mice, Aldh1l1-Cre/ERT2 (#031008, Jackson Laboratory) were crossed with Ai95-GCaMP6f mice to generate Aldh1l1-CreERT2; Ai95-GCaMP6f mice. GCaMP6 expression by CreERT2 recombination was induced by the administration of 1 mg of Z-4-hydroxy-tamoxifen (4-OHT, Tocris #3412) (0.2 mL, 20 mg/mL, i.p.) per mouse per day for three consecutive days.

### Surgery and sedation
Chronic cranial windows were implanted 1 week after the head bar surgery as described[55]. We used a 100 µm-thick glass coverslip over the barrel cortex for TPLSM (~3 mm²) and a 100 µm-thick polymethylpentene (PMP) coverslip over the entire neocortex for fUS experiments. A recovery period of 7–10 days minimum was respected before imaging experiments. During imaging sessions, mice were sedated with continuous perfusion of dexmedetomidine as follows: induction of anesthesia was done with 3% isoflurane and then decreased by 0.5% every 5 min down to 0% within 30 min, and dexmedetomidine was administered with a bolus (0.025 mg/kg s.c.) at the beginning of the session and an s.c. perfusion (0.1 mg/kg/h) during the entire session. Texas Red (70 kDa dextran, D1830, LifeTechnologies) or the pH-sensitive dye (see below) were administered intravenously (i.v.) with a retro-orbital injection prior to the decrease in isoflurane (under 3-2.5% isoflurane). Recordings started 20 min after the isoflurane cutoff. Body temperature was controlled with a rectal probe and maintained at 36.5 °C with a feedback-controlled heating pad. The animal was monitored throughout the imaging session using an infrared webcam (DCC3240N, Thorlabs).

### Hypercapnic and whisker stimulations
Hypercapnic and whisker stimulations were delivered with a home-made setup controlled with a custom Labview software (National Instruments). Mice received air supplemented with oxygen to reach a final concentration of 28% $O_2$ continuously through a nose cone. A valve, controlled electronically, was used to switch rapidly the mixture from air only to air and $CO_2$. The concentration of $CO_2$ at the mouse nostril was measured before each experiment to ensure that a 20% or a 10% plateau was reached within 2 s (SprintIR-R $CO_2$ sensor, Gas Sensing Solutions). Hypercapnic stimulations were repetitively applied with a minimum of 5 min interstimulus interval. Unilateral whisker deflections of the entire pad were achieved with a moving bar (5 Hz, 5 s). Both stimulations were triggered with the acquisition setup.

### Respiration measurement
Breathing was monitored during each imaging acquisition using a thermocouple placed in front of one nostril[37]. The respiratory rate (Hz) was extracted from the downward deflections due to inhalations.

### Arterial pressure and heart rate measurements
Blood pressure was assessed in a subgroup of four C57/BL6 male mice aged 6–7 months implanted with a telemetry transmitter device (PA-C10, Data Sciences) under isoflurane[56]. The catheter of the device was inserted in the left carotid artery. Mice were housed individually and a recovery period of at least 1 week was respected before the experiments. Hypercapnic stimulations were performed as described above under dexmedetomidine sedation and breathing rate was monitored. Blood pressure signals were sampled at 100 Hz. Systolic and diastolic arterial pressure were extracted with peaks and valleys detected using a custom Matlab script. The mean blood pressure was calculated as the mean value between each valley and consecutive peak. The heart rate was calculated by extracting the time between each peak (i.e. systolic events, corresponding to heart beats), and then converted to beats per minutes (bpm).

### Two-photon laser scanning microscopy
Imaging was performed using a femtosecond laser (Mai Tai eHP; SpectraPhysics) with a dispersion compensation module (Deepsee; SpectraPhysics) emitting 70-fs pulses at 80 MHz. Laser power was attenuated by an acoustic optical modulator (AA Optoelectronic, MT110-B50-A1.5-IR-Hk). Scanning was performed with Galvanometric scanner (GS) mirrors (8315KM60B; Cambridge Technology). AF488 and H-Ruby were excited at 880 nm and GCaMP6, GCaMP8, and Texas Red were excited at 920 nm. Emitted light was collected with either a LUMFLN60XW (Olympus, 1.1 NA) or a LUMPLFLN40XW (Olympus, 0.8NA) water immersion objective. Collected photons were sorted according to their wavelength using a dichroic mirror centered at 570 nm and the following filters: FF01-525/25 nm filter (Semrock) for AF488; FF01-609/62 for H-Ruby and FF01-620/20 for Texas Red. Photons were collected with 2 GaAsP (Hamamatsu) photomultipliers tubes. Customized LabView software was used to control the system.

### fUS imaging and data analysis
Cerebral blood volume (CBV) measurements were acquired using an ultrasound scanner (Iconeus One, Iconeus) as described previously[55,57]. Briefly, we used a linear ultrasound probe (128 elements, 15 MHz central frequency, Vermon) with an ultrasound sequence consisting of transmitting 11 different tilted plane waves (from −10° to 10° in 2° increments) with a 5500 Hz pulse repetition frequency (500 Hz frame rate of reconstructed images). Post-processing analysis was done in Matlab (version R2018a, MathWorks) with a custom-made software: the first 40 singular value decomposition (SVD) were removed and the power Doppler (PD) signal was further filtered with a Butterworth filter (fifth order). Data are shown as ΔPD/PD, with a baseline of 9 s (from 1 to 10 s). CBV flowing at low (10−30 Hz, 0.5–1.5 mm/s) and high (>60 Hz, >3 mm/s) axial velocities were separated by filtering as previously described[36]. In a subset of experiments, ultrasound localization microscopy (ULM) was used to measure the velocity of gas microbubbles injected i.v. (100 µL bolus, retro-orbital injection, Sonovue,

Braco) and recorded with a 3 min acquisition (8 V, superlocalization mode)[58].

## Functionalization of dextran with H-Ruby

H-Ruby dextran was obtained from the click reaction between H-Ruby-$N_3$ and a clickable dextran (70 kDa). The protocol for synthesis and characterizations can be found in Supplementary Informations. To a solution of H-Ruby-$N_3$ (2.5 mg, 4.1 μmol) and clickable dextran 70 kDa (30 mg, 0.3 μmol) in 400 μL degassed DMF was added copper sulfate pentahydrate (5 mg) and sodium ascorbate (5 mg) in water (100 μL). The reaction was stirred in the dark at 40 °C for 30 min. DMF was evaporated and the crude residue was dissolved in aq. 0.1 M EDTA (1 mL) to remove traces of cupper salt, purified over Sephadex® G-25 size exclusion column and lyophilized to obtain H-Ruby dextran conjugate as a pink solid (28 mg, molar ratio ~3 mol dye/mol dextran). A similar protocol was applied to obtain AF488 dextran conjugate, using AF488 azide (Lumiprobe GmbH, Hannover, Germany).

## pKa determination of H-Ruby Dextran conjugate

Fluorescence spectra of H-Ruby Dextran conjugate (excitation: 530 nm, 0.9 μM of fluorophore) were recorded using a FluoroMax-4 spectrofluorometer (Horiba Jobin Yvon) and pH-controlled buffers (ranging from pH 4.5 to pH 10.5) using a mixture of boric acid, citric acid and phosphate buffers according to the literature[59]. Fluorescence intensity at 580 nm was plotted against the pH and a Sigmoidal DoseResp fit provided the pKa value of 7.57 ± 0.18.

## pH measurements

H-Ruby dextran (70 kDa, 0.6 mg/kg) and AF488 dextran (70 kDa, 2 mg/kg) were co-administrated i.v. by retro-orbital injection. pH was calculated from the ratio of the average red and average green fluorescence signals in a region of interest (ROI) inside the vessel after subtraction of their respective backgrounds:

$$FR = \frac{F_{Red} - F_{RedBackground}}{F_{Green} - F_{GreenBackground}} \quad (M1)$$

pH calculation was performed as follows:

1. A look-up table was established by measuring R of H-Ruby-dextran and AF488 dextran in pH-controlled samples of fresh rat blood plasma. Data was fitted with a sigmoid function:

$$FR = \alpha_{system}\left(\frac{R_{Range}}{1 + e^{\alpha_{pH}(pH - pKa)}} + R_{Cte}\right) \quad (M2)$$

using nonlinear least square optimization for $R_{Cte}$, $\alpha_{pH}$ and $pKa$ in MatLab (see Supplementary Fig. 6). $\alpha_{system}$ is a constant specific to our microscope that characterizes the relative sensitivity of the system to the same amount of photons detected on the red channel or on the green channel. The following coefficients were obtained: $\alpha_{pH} = 2.64 \pm 2.1$, $pKa = 7.47 \pm 0.31$ and $R_{Cte} = 0.32 \pm 0.19$, $R^2 = 0.994$, error corresponding to 95 % confidence bound. Note that $R_{Cte}$ is not zero as a significant part of AF488 emission spectrum corresponds to the detection spectrum of our red channel.

2. As H-Ruby maximum fluorescence not only depends on pH but also on its chemical environment, depth in the brain, and optical properties of the tissue, $R_{Range}$ needs to be set for each recording. Therefore, $R_{Range}$ was calculated for each recording as:

$$R_{Range} = \left(\frac{FR_{Baseline}}{\alpha_{system}} - R_{Cte}\right)\left(1 + e^{\alpha_{pH}(7.4 - pKa)}\right) \quad (M3)$$

where $FR_{Baseline}$ is the average $FR$ in the baseline period when pH is assumed to be 7.4 in blood.

3. The pH was calculated as:

$$pH = pKa + \frac{\ln\left(\frac{R_{Range}}{\frac{FR}{\alpha_{system}} - R_{Cte}} - 1\right)}{\alpha_{pH}} \quad (M4)$$

## Image analysis

Diameter change in pial arterioles and RBC velocity in arterioles and capillaries were determined with line-scan acquisitions as described previously[18]. The LSPIV method was used to compute RBC velocity in arterioles and the angle algorithm for velocity in capillaries[60]. Diameter changes of penetrating arterioles were quantified by measuring their lumen area from 3 Hz movies with a custom Matlab script. GCaMPs fluorescence signals over chosen ROIs were computed as $\Delta F/F = (F - F_0) / (F_0)$, where $F_0$ represents the fluorescence of the baseline and F the fluorescence at time $t$. For the analysis of endothelial and smooth muscle cell GCaMP fluorescence, measurements required correction for dilation movements: ROIs (1/3 to 1/2 of the cells) were selected and each frame was realigned (cross-correlation) according to the ROI (See Supplementary Movie 1 for more details).

## Statistics and onset analysis

We computed our data in z score because of fluctuations in the baseline due to vasomotion[24]. Z score was calculated for each trial as follows: z score = (x − $\mu_{baseline}$) / $\sigma_{baseline}$ with a 9 s long baseline, with $\mu_{baseline}$ the mean of the baseline, and $\sigma_{baseline}$ the standard deviation (SD) of the baseline. Trials from the same vessel were averaged (with a 0.1 s interpolation) for analysis. As the mathematical functions underlying the responses to briefCO2 are currently unknown, model functions were first determined for each parameter measured (diameter, respiratory rate, RBC velocity, GCaMP fluorescence) using the average (all vessels) responses. Onsets were determined by fitting data from individual vessels with the specific model function and choosing the time to reach 10% of the peak of the fit. All fits were determined using the non-linear least mean square method for custom-made functions in Matlab. Details concerning the model fitting function determination for each parameter are available in Supplementary Table 1. Onsets of paired data and area under the curve of responses before and during $CO_2$ were analyzed using a non-parametric two-sided Wilcoxon rank sum test (GraphPad Prism, version 6). Data are represented as mean ± standard deviation.

## Reporting summary

Further information on research design is available in the Nature Portfolio Reporting Summary linked to this article.

## Data availability

The data generated in this study have been deposited as an Excel file on Zenodo and is publicly accessible at https://zenodo.org/records/11487983[61].

## Code availability

The codes used for the analyses are available from the corresponding author upon request. Matlab scripts used for onset determination are available online in the following GitHub repository: https://github.com/laboratory-charpak/NVC_and_CO2/tree/Dataset.

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

## Acknowledgements

We thank Manon Omnes for the management of mice colonies and performing some of the surgeries and Davide Boido for training M.T. to surgery and fUS experiments. We thank Kun Xie for his assistance during the blood pressure experiments. Financial support was provided to S.C. by the Institut national de la santé et de la recherche médicale (Inserm), the European Research Council (ERC-2013-AD6; 339513), the Fondation pour la Recherche Médicale (EQU201903007811), the Agence Nationale de la Recherche (NR-16-RHUS-0004 [RHU TRT_cSVD], CE37_2020_TF-fUS-CADASIL and CE14-0026-01_Brain-mapping-of-pH-and-CO2), the Fondation Leducq Transatlantic Networks of Excellence program (16CVD05, Understanding the role of the perivascular space in cerebral small vessel disease), the Fondation Alzheimer France (M21JRCN009) and the IHU FOReSIGHT [ANR-18-IAHU-0001] supported by French state funds managed by the Agence Nationale de la Recherche within the Investissements d'Avenir program, and to A.J. by the National Institutes of Health, USA (1RF1NS128963). M.T. was supported by a fellowship from the Fondation pour la Recherche Médicale (SPF201909009103).

## Author contributions

M.T. and S.C. designed the study, interpreted the data and wrote the manuscript. M.T. performed and analyzed all experiments. A.J. participated to the conception of the project and the interpretation of the data. E.C. developed the onset measurement with fitting and the theoretical approach and software for pH measurements. A.-K.A. developed some scripts for fUS and two-photon analysis. Y.G.H. maintained the performances of our custom-built microscope and participated to set rapid and controlled $CO_2$ application system. S.P. and M.C. synthetized the pH sensors. P.O. and J.F. prepared the animals and helped with blood pressure and heart rate measurements. All authors edited the manuscript.

## Competing interests

The authors declare no competing interests.
