## [Peer Review File · Nature Communications]

Neurovascular coupling and CO₂ interrogate distinct vascular regulationsREVIEWER COMMENTS

Reviewer #1 (Remarks to the Author):

In this study, the authors Tournissac M et al. explore whether systemic CO₂ from inhalation affects neurovascular coupling (NVC). The main idea is to determine if the CO₂ generated by neuronal metabolism, contributes to feedforward NVC, or if NVC is instead mediated by canonical transmitters and vasoactive messengers without the involvement of CO₂. This is a timely manuscript given a recent paper by Hosford et al. showing that peripheral CO₂ occludes NVC to functional brain activation. Employing a combination of two-photon imaging in the barrel cortex and functional ultrasound for both regional and whole brain measurements, the authors specifically examine the onsets of local changes in arteriole diameter and RBC velocity at different points in the vascular tree (surface pial vessels, penetrating arterioles, and the capillary bed) as well as neurovascular unit pH and intracellular calcium signals. The experiments involve brief exposure to high % CO₂ and whisker stimulation. The findings reveal that brief CO₂ exposure leads to reversible acidification of all cells in the neurovascular unit occurring ~3 seconds prior to a transient arteriole dilation. The authors use this time lag to study if NVC, triggered by whisker stimulation, is affected or not by the CO₂ induced acidification. Importantly, the persistence of NVC under continuous CO₂ inflow leads the authors to conclude that CO₂ does not play a role in NVC. The investigation of the carotid artery to explain the drop in RBC velocity was nicely done. The study is well conducted, interesting, timely, and uses a combination of powerful techniques. Yet, while the authors have a consistent message on the lack of effect of CO₂ on NVC, I would interpret some of the effects differently. Furthermore, proving that the two pathways do not interact is challenging because a lot depends on the specifics. I also find that the application of the CO₂ and whisker stim protocols are inconsistent across vessel segments. Finally, it is also disappointing that no cellular mechanisms are explored, which makes the study somewhat superficial, as mechanistic data could help support the authors claims. Nevertheless, it is an interesting and timely paper using state of the art methods. Please see my specific comments.

Major

1) There are two ways to think about the timing protocol of NVC plus CO₂ in Fig 4a-d. First, in the authors description, they are attempting to see if the NVC response is affected by the early acidification caused by CO₂ before the CO₂ dilation takes place, this gives them only 2-3 seconds to see if NVC is affected or not. Indeed, the NVC response looks normal, so they conclude that NVC and CO₂ do not interact. However, this only shows that NVC is insensitive to the early acidification itself, before the acidification has reached its peak. It is possible that 1) this early acidification is not strong enough to effect NVC and the peak acidification might be, or 2) that acidification itself is not actually necessary for the CO₂ dilation and the authors are testing the wrong component of the CO₂ response. For example, it is notable that it takes 2-3 seconds of acidification before the vessel starts to dilate to CO₂. This means that either the early acidification is insufficient to dilate the arteriole, or that acidification is not the variable that dilates the arteriole. Either way, it might be expected that NVC is insensitive here, but this does not mean the pathways do not interact. Indeed, the second way to think about this experiment is an occlusion test. If the two challenges converge on the same cell pathway or same mediator (CO₂), one might expect (if certain conditions are met) that the individual responses are not additive. For example, if the NVC response causes a 10% dilation and the CO₂ response is 10%, then combining them should give you a response of 10%, if they "occlude" each other. However, if they worked through different cell pathways/mediators, the response could add, as long as a ceiling effect didn't occur in the vessel and the pathways were sufficiently activated (more on this in another point). Here, the combined response would be 20% (no occlusion). The NVC response and separately the CO₂ response, each take about ~17sec to rise and fall, and they have staggered the two challenges only by 2 seconds, so there is still lots of time for these pathways to interact. From this perspective, it appears that the two challenges do occlude each other; whether the two challenges are staggered or not, the peak amplitudes are the same. To me, this data shows that the slightly later CO₂ dilation in the grey trace (the second peak) is occluded by NVC, otherwise the trace should have bumped up towards 10 SD for this second peak. Alternatively, the pink trace would be 10 SD if the were separate and additive, because the response amplitude of NVC alone is ~5SD,

the response amplitude of CO₂ alone is ~5SD, and amplitude of both combined is still ~5SD. Yet, the authors are arguing the opposite, that the NVC dilations occur through separate mechanisms to CO₂-mediated dilations. I think it is only safe to say that the early acidification does not occlude the NVC response.

An occlusion experiment could be more clearly tested if the timing of the dilations were set to be simultaneous. Here, stimulate the whiskers 4 seconds into the CO₂ challenge. This will align the dilations and one can more clearly see if they occlude or not. However, I do not think this experiment is needed because the data already basically shows this: they occlude.

However, the author's conclusion, that the pathways appear separate and do not interact, does make sense in other protocols at different parts in the vascular tree. For example, when the CO₂ challenge is continuous at the capillary bed, whisker stimulation produces an unabated increase in RBC velocity. Here, it would seem the pathways do not occlude each other, but, is this because the capillary response is different than the arteriole response, or because the protocol is different? More on this last point below. In the end I am left confused. I see occlusion in some conditions and a lack of occlusion in others. It is possible that there are separate mechanisms in certain vessel segments and there is convergence of mechanisms in other vessel segments. This point must be resolved, and the interpretations clarified.

2) I do not understand why at the penetrator the authors try to time the arrival of two transient response, but at the capillary they use constant CO₂ and add NVC on top? The authors need to use the same protocol at the penetrator and capillary level, with 2P and with fUS. My suggestion is to use both protocols consistently across conditions. Try for the timed arrival of the two transient challenges in each vessel segment of interest, and separately see if the amplitude of the NVC response is different in the presence of constant CO₂. The authors should do this in the vascular segments of interest (penetrator and capillary) in 2P and in fUS.

3) Another point about testing whether these pathways interact or not. The authors may need to hit the top of the dose response curve on one pathway and then see if there is no further effect from the other pathway. For example, if two pathways converge onto the same mediator or mechanism, if each pathway only drives the pathway at 25% of maximal, then why would not the two pathways combine to reach 50% of max? However, if one pathway is topped out at 100% of what it can achieve, then the other pathway should have little effect if there is convergence or interaction. This does get tricky however because the vessel needs to not reach its ceiling from one pathway being maxed out, and also, we know that NVC has several redundant pathways, so perhaps only some mechanisms interact with CO₂. Given this, the best strategy might be to see how a 20% transient CO₂ response behaves when it coincides with a maximal NVC response. There is little indication in the paper for how strong of an NVC response the authors are using. From my perspective, looking at the one place they show % change, this is a weak functional hyperemia response, as it's a 10% diameter increase at most. Functional hyperemia can be as large as 40% in an awake mouse, and in an anesthetized prep could be closer to 20%. This might set the occlusion/interaction experiment up to fail right from the start as the NVC response is simply too weak.

4) When the CO₂ challenge is continuous, what does the pH look like in the neurovascular unit? The authors primarily use a continuous CO₂ protocol to test their hypothesis, but only measured the pH to transient CO₂. Perhaps the acidification gets more pronounced or recovers back to baseline when the CO₂ application is sustained. Depending on what pH does, this would change the interpretation of the occlusion test in this condition. This is important to test.

5) Whether the NVC and CO₂ pathways occlude or not would be better demonstrated by showing they have different mechanisms. If the effects do not occlude, the authors could show that NVC is sensitive to certain blockers that CO₂ is not sensitive to, and that CO₂ is sensitive to certain blockers that do not affect NVC. I am aware of previous literature showing that CO₂-induced vasodilations are sensitive to compounds that affect NVC, such as COX inhibitors and NOS inhibitors, but this becomes difficult when considering direct mediator vs modulator effects, especially with NO. Nevertheless, are there blockers that can select for one mechanism over the other? If so, this would strengthen the authors argument.

If the pathways do occlude in some condition, such as at the penetrating arteriole, then it would be fitting to test the mechanism proposed by the Hosford paper. Use an AAV to knockdown the NaHO3 co-transporter in astrocytes and see if this protein is involved in NVC or not or not. I recognize that this is not a trivial experiment to do. Perhaps the authors could try this or the pharmacology tests described above? I am not saying the authors must have manipulations like this, but it would strengthen their arguments and this paper considerably.

6) The sedative dexmedetomidine has both blood pressure and respiratory effects. Blood pressure first raises above normal followed by a decrease below normal values. Respiration is also depressed in the drug. Do the authors know what phase their experiments were in: hypotensive vs hypertensive? This caveat should at least be discussed.

On this note, if the aim here is to test the same hypothesis as the Hosford paper (but with different imaging modalities), the authors here should make sure the anesthetic/sedative is the same. It is expected that every anesthetic or sedative has some impact on cerebral blood flow and its control. Looking it up, that paper used α -chloralose. The authors here should test their main effect in α -chloralose to rule out this caveat.

Minor

7) I need clarification on the z score approach. The following formula is given: $z \text{ score} = (x - \mu_{\text{baseline}}) / \sigma_{\text{baseline}}$. μ is not defined but I assume it is the mean of the baseline values. Is that correct? The authors report "SD" as their primary measure in most of their graphs. Is it truly standard deviation that the authors are plotting here or should it be better labelled as "z score". Plotting stand deviation over time makes little sense to me.

8) It would be nice to see what the Alexa488 signal and H-Ruby signal look like separately and as a ratio.

9) In fig 4g and fig 5e, where the % change is plotted in constant CO₂, I am confused what is being plotted here exactly. Is this the baseline increase to CO₂? Or is it the evoked NVC response from the new baseline in CO₂? I assume the latter, and if so the connecting line between the points is confusing.

10) Line 76 "which enabled us to uncover"

11) Line 112, "imposes" should probably be "implies"

Reviewer #2 (Remarks to the Author):

This is a very well written and carefully conducted study investigating the role of CO₂ in neurovascular coupling responses. CO₂ has a potent vasodilatory effect, but it is controversial as to whether normal production of CO₂ during brain metabolism is a dilatory signal to brain arterioles. The authors point out that many prior CO₂ inhalation studies used a "sledge hammer" approach with long duration, high level CO₂ (minutes) and anesthetized preparations. In this respect, the authors take an innovative new angle using different animal preparations (chronic windows, sedation) and brief delivery of CO₂ (carefully timed short duration CO₂ at time-scales relevant to neurovascular coupling). The key experiment determines whether brief CO₂ inhalation has any effect on whisker-induced neurovascular coupling responses. In contrast to prior studies, the authors find no evidence that CO₂ inhalation affects neurovascular coupling. Overall, I think this is a very elegant study and I concur with most of the presented findings. However, there are some measurements that remain difficult for me to interpret and could use further clarification/analysis.

1) The authors show an interesting decrease in average GCAMP6/8 fluorescence across all neurovascular cells and attribute this to pH sensitivity of the calcium sensor and CO₂-mediated

acidosis. However, the signal change may reflect some real intracellular calcium change in the cell. For example, with smooth muscle cells, a decreased in calcium is consistent with vasodilation. How do the authors rule out the possibility that GCaMP signal change is dependent upon both calcium change and acidosis in their studies?

2) If GCaMP6/8 signal does encode some information about cellular calcium, then the analysis of the fluorescence signals may be too simplistic. Calcium signal patterns in endothelial cells, astrocytes, and neurons are spatiotemporally complex. For example, endothelial cells have a variety a different calcium event types (<https://www.science.org/doi/10.1126/sciadv.abh0101>). The meaning of an average signal decrease is not obvious, especially for cells such as endothelium and astrocytes. Even for neurons, one wonders if decreased neuropil calcium is indicative of decreased pyramidal neuron firing.

3) Presumably, the level and duration of CO₂ should not cause any swelling of brain tissue. A confirmation of this would be good to have since shifting of the focal plane by change in brain size may affect GCaMP6/8 fluorescence intensity.

4) The H-Ruby dextran is an innovative idea. If I understand correctly, H-Ruby fluorescence increases with greater acidosis. However, fluorescence of an inert dye can also increase as vessels dilate, and this is particularly evident with approaches using broad integration of signal in the axial dimension (<https://www.nature.com/articles/s41467-020-19851-1>). Appropriately, the authors use a ratiometric approach with AF488 as a concurrent signal in a second channel. But no raw data is shown on how these two signals behave with respect to each other. Can the authors please show raw data traces? Can they be certain that all the H-Ruby signal change is solely due to pH fluctuation and not luminal volume change?

5) Fig. 4d. There is a delayed second peak in the vasodilatory response of control mice. Is this a consistent finding in the author's stimulation paradigm? If so, there seems to be a deviation between experimental groups at this phase, and a separate analysis of this phase may be warranted.

6) The authors contrast their findings with the recent study by Hosford et al. It seems to me that a major difference is the magnitude of resting state CBF change induced by CO₂ in the current study and that of Hosford et al. Hosford et al. show a ~2-fold increase in resting state CBF change (ASL-MRI) in their paradigm, and the current study shows ~1.3-fold change in velocity even with prolonged CO₂ (Fig. 4g). If prolonged, high level CO₂ maximally dilates arterioles then there is no longer a reserve for neuronally-evoked dilation. In contrast, the current study uses a more finessed approach that preserves some room for further arteriolar dilation. It is actually hard to make a true head-to-head comparison of how CO₂ in the two studies is changing resting state CBF with the data provided. Can the authors comment on whether this conclusion is valid?

Minor

- Fig. 4e. Please provide a more in depth explanation of the "occlusion hypothesis". The schematic may be too simplified for readers to quickly understand this concept.

- Figure 1,2,3, in the time-course panels, please clarify what the downward pointing arrows mean. I presume it means the point of significant change from baseline, but this is not specified in the legend.

- Results: In the calcium imaging studies, it is not entirely clear that the varied NVU cell types imaged were experiments in different types of mice. This could be made more clear in the Results section, as readers may think that different NVU cells are being measured concurrently, raising concern of contaminating signals between cells since all the GCaMP6/8 traces look similar.

Reviewer #3 (Remarks to the Author):

The main topic of this manuscript is outside my expertise. Based on the editor's request, I hereby provide my evaluation of the new pH probe used in the study.

The authors Tournissac et al. describe in their work the use of a pH responsive rhodamine probe H-Ruby dextran. Design and synthesis of the probe is based on published work (properly cited). The major difference when compared to the previous work is that here the authors used Rhodamine B instead of X rhodamine as the core fluorophore structure. This difference is reflected in slightly different photophysical properties, which however, do not significantly affect the probe's performance. The pH responsive characteristic of the new probe was studied. The probe responds well to pH changes in the range of 10 - 4.5, which is relevant to the study. The synthesis is adequately described in the supporting information. The only missing information in the supplementary information is that the clickable dextran is not labeled in the scheme as compound Nr. 6. Similarly, the click product should be depicted as compound Nr. 7.

Based on the available data, I conclude that the use of the probe is appropriate, even though the design and the synthesis is based on known principles and strategies. All necessary details, such as the synthetic part and pH response evaluation are sufficiently described in the article.

Reviewer #4 (Remarks to the Author):

The study conducted by Tournissac et al. addresses the fundamental question of CO₂'s role in neurovascular coupling. The authors employed a combination of functional ultrasound and two-photon imaging with pH sensors in the mouse barrel cortex. They proposed an innovative method involving whisker stimulus and hypercapnic challenges to analyze CO₂'s impact.

The findings revealed an initial decrease in blood velocity during a brief CO₂ challenge, followed by a diameter increase a few seconds later. The authors hypothesized that this initial decrease is linked to upstream flow modifications, as indicated by a drop in cerebral blood volume (CBV) observed with functional ultrasound in the internal carotid.

The CO₂ challenge yielded no impact on the response to whisker stimulation leading to the conclusion that CO₂ has no role in the neurovascular coupling.

The article is well-written, with clear research questions and a detailed description of the methodology.

However, I have the following comments that I would like to be addressed:

The relevance of the CO₂ challenge as a model for CO₂ release from local brain activity is unclear. Figure 3 partly addresses this but lacks specific results. Without quantitative units in fluorescence measurements, it's difficult to compare the CO₂ concentrations during brief CO₂ challenges and whisker stimulation (WS). The study lacks an investigation of pH during WS. A quantitative comparison of these curves is essential to justify using the CO₂ challenge as a model for CO₂ release.

While the authors investigated CBV in the internal carotid, they did not examine other systemic cardiovascular effects, such as heart rate and blood pressure, which could influence brain flow during the hypercapnic challenge. They also did not discuss how autoregulation might affect flow under these conditions.

The role of ULM images in the study is unclear. The authors quantified blood velocity in the internal carotid, but it's not evident how this data was used in the study. Were these measurements solely for positioning CBV measurements with functional ultrasound?

Statistically, a non-significant difference is not equivalent to no difference. How do the authors

address this? Additionally, Figure 4h shows some early differences in neuronal response (overshoot) that are not visible when testing area under the curve (AUC) values.

Minor comments:

Figure 2, a: Please include diameter measurements in micrometers.

Figure 5, a-b: Why is the y-axis scale bar different for high and low velocities? Functional ultrasound cannot detect an increase in slow vascular flows, although two-photon experiments are averaged for n=44 vessels from 25 mice, compared to n=5 mice in functional ultrasound.

Line 209: The term "fast CBV" is unclear.

Line 221: The use of commas seems unnecessary.

Responses to Reviewer #1:

General comments:

In this study, the authors Tournissac M et al. explore whether systemic CO₂ from inhalation affects neurovascular coupling (NVC). The main idea is to determine if the CO₂ generated by neuronal metabolism, contributes to feedforward NVC, or if NVC is instead mediated by canonical transmitters and vasoactive messengers without the involvement of CO₂. This is a timely manuscript given a recent paper by Hosford et al. showing that peripheral CO₂ occludes NVC to functional brain activation. Employing a combination of two-photon imaging in the barrel cortex and functional ultrasound for both regional and whole brain measurements, the authors specifically examine the onsets of local changes in arteriole diameter and RBC velocity at different points in the vascular tree (surface pial vessels, penetrating arterioles, and the capillary bed) as well as neurovascular unit pH and intracellular calcium signals. The experiments involve brief exposure to high % CO₂ and whisker stimulation. The findings reveal that brief CO₂ exposure leads to reversible acidification of all cells in the neurovascular unit occurring ~3 seconds prior to a transient arteriole dilation. The authors use this time lag to study if NVC, triggered by whisker stimulation, is affected or not by the CO₂ induced acidification. Importantly, the persistence of NVC under continuous CO₂ inflow leads the authors to conclude that CO₂ does not play a role in NVC. The investigation of the carotid artery to explain the drop in RBC velocity was nicely done. The study is well conducted, interesting, timely, and uses a combination of powerful techniques. Yet, while the authors have a consistent message on the lack of effect of CO₂ on NVC, I would interpret some of the effects differently. Furthermore, proving that the two pathways do not interact is challenging because a lot depends on the specifics. I also find that the application of the CO₂ and whisker stim protocols are inconsistent across vessel segments. Finally, it is also disappointing that no cellular mechanisms are explored, which makes the study somewhat superficial, as mechanistic data could help support the authors claims. Nevertheless, it is an interesting and timely paper using state of the art methods. Please see my specific comments.

We thank Reviewer #1 for his overall positive general comments and respond below to her/his interrogations concerning our experimental protocols and their results.

Major comments:

1) There are two ways to think about the timing protocol of NVC plus CO₂ in Fig 4a-d. First, in the authors description, they are attempting to see if the NVC response is affected by the early acidification caused by CO₂ before the CO₂ dilation takes place, this gives them only 2-3 seconds to see if NVC is affected or not. Indeed, the NVC response looks normal, so they conclude that NVC and CO₂ do not interact. However, this only shows that NVC is insensitive to the early acidification itself, before the acidification has reached its peak. It is possible that 1) this early acidification is not strong enough to effect NVC and the peak acidification might be, or 2) that acidification itself is not actually necessary for the CO₂ dilation and the authors are testing the wrong component of the CO₂ response. For example, it is notable that it takes 2-3 seconds of acidification before the vessel starts to dilate to CO₂. This means that either the early acidification is insufficient to dilate the arteriole, or that acidification is not the variable that dilates the arteriole. Either way, it might be expected that NVC is insensitive here, but this does not mean the pathways do not interact. Indeed, the second way to think about this experiment is an occlusion test. If the two challenges converge on the same cell pathway or same mediator (CO₂), one might expect (if certain conditions are met) that the individual

responses are not additive. For example, if the NVC response causes a 10% dilation and the CO₂ response is 10%, then combining them should give you a response of 10%, if they “occlude” each other. However, if they worked through different cell pathways/mediators, the response could add, as long as a ceiling effect didn’t occur in the vessel and the pathways were sufficiently activated (more on this in another point). Here, the combined response would be 20% (no occlusion). The NVC response and separately the CO₂ response, each take about ~17sec to rise and fall, and they have staggered the two challenges only by 2 seconds, so there is still lots of time for these pathways to interact. From this perspective, it appears that the two challenges do occlude each other; whether the two challenges are staggered or not, the peak amplitudes are the same. To me, this data shows that the slightly later CO₂ dilation in the grey trace (the second peak) is occluded by NVC, otherwise the trace should have bumped up towards 10 SD for this second peak. Alternatively, the pink trace would be 10 SD if they were separate and additive, because the response amplitude of NVC alone is ~5SD, the response amplitude of CO₂ alone is ~5SD, and amplitude of both combined is still ~5SD. Yet, the authors are arguing the opposite, that the NVC dilations occur through separate mechanisms to CO₂-mediated dilations. I think it is only safe to say that the early acidification does not occlude the NVC response.

An occlusion experiment could be more clearly tested if the timing of the dilations were set to be simultaneous. Here, stimulate the whiskers 4 seconds into the CO₂ challenge. This will align the dilations and one can more clearly see if they occlude or not. However, I do not think this experiment is needed because the data already basically shows this: they occlude. However, the author’s conclusion, that the pathways appear separate and do not interact, does make sense in other protocols at different parts in the vascular tree. For example, when the CO₂ challenge is continuous at the capillary bed, whisker stimulation produces an unabated increase in RBC velocity. Here, it would seem the pathways do not occlude each other, but, is this because the capillary response is different than the arteriole response, or because the protocol is different? More on this last point below. In the end I am left confused. I see occlusion in some conditions and a lack of occlusion in others. It is possible that there are separate mechanisms in certain vessel segments and there is convergence of mechanisms in other vessel segments. This point must be resolved, and the interpretations clarified.

This long main comment requires that we first clarify 2 points before answering to the issues raised. 1) In our Results section, we never used the word occlusion but once (line 192) in brackets. The reason is that our data show that we never reach any type of "occlusion" or "saturation" with our experimental protocols. Instead, we have clearly observed additivity between all responses. This point, which was not clear in the initial manuscript, is now clarified in the new Figure 4 (and in Supplementary Fig4).

2) We believe that it is important to define NVC. It involves several signaling pathways which dynamics vary with the strength of neuronal activation. It is now well accepted that NVC onset occurs within ~1-2 seconds in rodents with a time-to-peak of about ~1-2s. This defines what was initially considered as the timing of NVC in awake mice, the time to peak being slower during anesthesia¹. It is also established that even for a brief stimulation there are delayed responses that involve astrocytes^{2,3}. The recent paper by Institoris et al. (2022) nicely demonstrated that clamping calcium in astrocytes decreases the delayed vascular response. In our work, we have principally investigated the role of CO₂ on the initial component of NVC, the goal being to determine whether CO₂ is necessary or not to trigger NVC (i.e., whether CO₂ is a mediator of NVC or not). It is now more clearly specified in the Results (lines 159-160, 165, 183-184) and the Discussion (lines 264-7, 277).

We thank Reviewer #1 for asking us to clarify the results obtained with our experimental protocol based on "paired" stimulations, and to show what level of additivity occurs between NVC and CO₂ responses (former Fig.4a-d). This was clearly missing in the initial manuscript and for an obscure reason, we previously considered that it would blur our message. We now present all the steps and results that we used to build Fig. 4 (see below), and which show full additivity of responses to CO₂ and NVC (i.e. the absence of occlusion). Now, we first show the responses to NVC and briefCO₂ that were done separately at the beginning of each experiment, as controls (Fig.4e). We then compare the calculated summation of these separated responses with the experimental summation of the responses to both stimuli (Fig. 4f-h and Supplementary Fig.4), the WS occurring either simultaneously with briefCO₂ or with a 2 s delay. The conclusion is that 1) the calculated summation of separate responses and the experimental summation with our first paradigm (simultaneous stimulations) are similar when considering the first 5-6 s of NVC ("1st phase", up to the response peak). This means that there is a perfect additivity and no occlusion of both responses during this period. 2) The first 5-6 s of NVC are also similar when WS is delayed by 2 s (second paradigm). Thus, NVC remains constant during the early period of CO₂ diffusion/acidosis of the NVU, indicating that CO₂/pH changes do not influence the triggering of NVC.

After the peak ("2nd phase"), 1) there are differences between calculated and experimental responses (secondary humps in the black and orange curves in Fig. 4f and g, highlighted in Supplementary Fig. 4c) for both paradigms, and 2) the secondary "hump" ("2nd phase") seems larger when stimulations occur simultaneously (first paradigm). We verified the hypothesis by measuring, for each experiment, the difference between calculated and experimental summations. The averaged differences observed with the two stimulation paradigms are not significantly different (Supplementary Fig 4c) with our small dataset (note that the SD is larger during the late phase than during the early phase of NVC). It could become (or not) significant with more experiments. We did not add such experiments because they are out of the topic of the present work which is to investigate the effect of CO₂ / acidosis on the triggering of NVC. Still, if a certain level of acidosis truly decreased the late phase, it could result from multiple reasons that we briefly mention in the Discussion:

- 1) During simultaneous stimulation, the dilation due to NVC could boost the delayed effect of CO₂ enhancing the secondary hump. The effect could be purely mechanical, with a displacement on the length-tension curve of SMCs due to a change of overlap between actin and myosin filaments. The boost would be smaller when NVC is delayed by 2 s.
- 2) Many other hypotheses could be raised, such as an effect of CO₂ on the mechanisms underlying the negative undershoot that appears in control condition at the end of NVC.

In Results: "Fig. 4c shows that neuronal responses (perivascular bulk calcium; Thy1-GCaMP6s mice) were not affected during CO₂ diffusion and acidification of the periarteriolar column. Note that the drop of GCaMP6 fluorescence following neuronal activation resulted from CO₂-induced acidosis and not from a delayed inhibition. In addition, the superposition of arteriole dilations in both conditions show that the first phase of dilation (the first 5-6 s leading to the dilation peak) was similarly not affected by CO₂ (Fig. 4d). To analyze the extent to which NVC and CO₂ vascular responses were additive or not, we compared the calculated summation of these responses measured separately (Fig. 4e) with the experimental responses to both stimuli (i.e. experimental summation), with whisker stimulation occurring either simultaneously with briefCO₂ (Fig. 4f) or with a 2 s delay (Fig. 4g). Fig. 4f-h shows that the calculated summation of separate responses and the experimental summation with the two stimulation paradigms are similar when considering the first 5-6 s of NVC (up to the response peak). This means that there is perfect additivity and no occlusion of the two responses during this period. Thus, the early period of CO₂ diffusion and acidification of the NVU does not influence the initiation of NVC, although it may influence the second phase of NVC (see Supplementary Figure 4 and Discussion)."

In Discussion (lines 281-289): “However, NVC clearly occurred during acidosis and CO₂ diffusion: NVC and CO₂ responses showed full additivity during the first 6 seconds of NVC. A delayed second phase (or hump) appears during "paired" stimulation experiments. The shape of the hump depends on the timing of the summation. However, the difference between experimental and calculated summation of both NVC and CO₂ responses suggests a possible modulation of this late phase by acidosis (Supplementary Figure 4c). Although the investigation of this possible modulation is beyond the scope of our study, many mechanisms could be hypothesized, such as the degree of actin/myosin filament overlap in SMCs at the onset of second dilation or an effect of CO₂ on the mechanisms underlying the undershoot that terminates the NVC response.”

To conclude:

-The 20% CO₂ stimulation for 10 s is very strong and modifies vascular pH by ~0.15 at its peak. The peak of NVC occurs within ~7 s of briefCO₂ stimulation (Fig.4d-g), with a time to peak from onset of whisker stimulation of ~5-6 s. The peak of NVC thus occurs when pH decreased by ~0.12 unit (from Fig 3d, right). This indicates that the progressive decrease of pH does not affect the initiation (first 5-6 s) of NVC. As pH (and not CO₂) is a well-accepted parameter that modulates arteriole diameter⁴, we believe that these 5-6 s are very indicative of the lack of effect of CO₂ diffusion/acidosis on the initial component of NVC, even though the acidosis peak was not reached. Nevertheless, we acknowledge that some cellular consequences of this acidification may require more time and it is the reason why we directly investigated NVC under a 10 min-long CO₂ diffusion/ acidification (Fig. 4-5), instead of testing NVC during the entire course of the acidosis due to briefCO₂. This is clarified in the Discussion section.

-Because 1) there is a perfect additivity of NVC and CO₂ during the initial phase of NVC, 2) we don't have the tools to determine precisely whether CO₂ and NVC work through pathways that converge early or late in the dilation process, 3) the CO₂ diffusion/acidosis could affect all neuronal/glial/NVU mechanisms implicated in the delayed phase of NVC, we consider that investigating the mechanisms underlying the delayed phase of coupling is beyond the scope of the study. This point is now discussed further as follows:

“In rodents, NVC onset occurs within ~1-2 s with an initial response peaking at about ~1-2 s (for brief stimulations) and, depending on the stimulation strength, a delayed response that involves astrocytes^{2,3}. In our work, we have principally investigated the role of CO₂ on the initial component of NVC, the goal being to determine whether CO₂ is a necessary mediator to trigger NVC. (...) Although the investigation of this possible modulation is beyond the scope of our study, many mechanisms could be hypothesized, such as the degree of actin/myosin filament overlap in SMCs at the onset of second dilation or an effect of CO₂ on the mechanisms underlying the undershoot that terminates the NVC response.”

2) I do not understand why at the penetrator the authors try to time the arrival of two transient response, but at the capillary they use constant CO₂ and add NVC on top? The authors need to use the same protocol at the penetrator and capillary level, with 2P and with fUS. My suggestion is to use both protocols consistently across conditions. Try for the timed arrival of the two transient challenges in each vessel segment of interest, and separately see if the amplitude of the NVC response is different in the presence of constant CO₂. The authors should do this in the vascular segments of interest (penetrator and capillary) in 2P and in fUS.

We have partially responded to this remark in our response to comment 1. We summarize our responses to comment 2:

1) The "paired" protocol at the penetrating arteriole site aimed at generating a brief and strong diffusion of CO₂ and an acidosis of the arteriole NVU. The data show that under these conditions there was no occlusion and full additivity (new Fig4 i-g) for the first part of NVC.

2) The second paradigm, at the capillary level, had a larger aim: it was to show that under Hosford et al. conditions (10%, 14 min, although with different anesthetics, see below) and after stabilization of resting flow, the percentage change in RBC velocity to WS remains the same. We consider that the capillary response reflects both the local vascular response to neuronal activation and the consequences of the vascular conducted response to upstream arterioles and sphincters (so both capillary and arteriole compartments).

4) We want to stress again that our unique fUS analysis allowing to discriminate between high and low speed CBV, corresponding to both arteries/veins versus capillary bed compartments (up to now, no other lab has previously shown such subtle analysis⁵), shows that the change of power doppler is unchanged under continuous CO₂ in both compartments. Note that, unfortunately, this technique does not allow to quantify resting values of CBV but only CBV changes.

5) To convince Reviewer #1 that these capillary RBC velocity measurements (with 2P or fUS) also encompass what occurs at the arteriole level, we added new experiments: we simultaneously recorded diameter and RBC changes to whisker stimulation (with 2P and line-scan acquisitions) to extract blood flow, which corresponds better to CBV changes. Similarly to capillaries and fUS experiments, during continuous CO₂ stimulation (10%, 14 min), NVC remained unchanged in pial arteries, even if resting blood flow increased by 60% (see Fig 4o-s, and Results section lines 186-195).

3) Another point about testing whether these pathways interact or not. The authors may need to hit the top of the dose response curve on one pathway and then see if there is no further effect from the other pathway. For example, if two pathways converge onto the same mediator or mechanism, if each pathway only drives the pathway at 25% of maximal, then why would not the two pathways combine to reach 50% of max? However, if one pathway is topped out at 100% of what it can achieve, then the other pathway should have little effect if there is convergence or interaction. This does get tricky however because the vessel needs to not reach its ceiling from one pathway being maxed out, and also, we know that NVC has several redundant pathways, so perhaps only some mechanisms interact with CO₂. Given this, the best strategy might be to see how a 20% transient CO₂ response behaves when it coincides with a maximal NVC response. There is little indication in the paper for how strong of an NVC response the authors are using. From my perspective, looking at the one place they show % change, this is a weak functional hyperemia response, as it's a 10% diameter increase at most. Functional hyperemia can be as large as 40% in an awake mouse, and in an anesthetized prep could be closer to 20%. This might set the occlusion/interaction experiment up to fail right from the start as the NVC response is simply too weak.

We believe that the Reviewer has been confused by the fact that our results and our conclusion were not optimally presented (see comment 1). There is no sign of occlusion in our data and we never reach the ceiling of dilation. It is to note that in Hosford et al. the addition of acetazolamide on top of the 10% CO₂ stimulation further increased blood flow up by a factor~3, indicating that even under their condition, maximal dilation was not reached.

Again, our goal is not to use the "paired" protocol to decipher where the two signaling pathways interact but to use the timing of briefCO₂ to test the hypothesis that exogenous CO₂ would alter NVC, as CO₂ diffusion and acidification of the periarteriolar column precedes arteriole dilation by several seconds. In these conditions, there is no sign of occlusion. We don't claim more. By definition, increasing resting diameter near maximal dilation with one of the stimuli may affect the other response but this is not what we want to demonstrate. The fact that NVC remains under continuous CO₂ allows to conclude that such protocol is not valid to state that CO₂ diffusion/pH is a major component of NVC. This is particularly important in view that both neurovascular coupling and continuous CO₂ are used in human imaging, freely breathing.

In addition, a 10 % change of arteriole diameter is not so small, for a physiological stimulation lasting 5 s. Please note that in Fig. 2b, we are showing the response to briefCO₂ in % change, and not to WS. We are now showing the dilation to WS in % change in pial arterioles in Fig. 4p. In addition, the Figure below shows a similar % change between penetrating and pial arterioles, peaking around 10-12% dilation. Importantly, it corresponds to what has been published in awake animals with a 5 s air puff or whisker brushing in several publications, ranging from 10 to 15% increase in diameter^{2,6-8}, including our work with a 5 s single whisker deflection¹. Of course, this value will depend on the stimulus intensity, the brain state, the anesthetic used or the anesthesia depth.

4) *When the CO₂ challenge is continuous, what does the pH look like in the neurovascular unit? The authors primarily use a continuous CO₂ protocol to test their hypothesis, but only measured the pH to transient CO₂. Perhaps the acidification gets more pronounced or recovers back to baseline when the CO₂ application is sustained. Depending on what pH does, this would change the interpretation of the occlusion test in this condition. This is important to test.*

We totally agree. The hypothesis that intravascular pH varies during continuous CO₂, and does not reach a steady state is important to test as it could underly the absence of effects of CO₂ on NVC. We thus performed new experiments and measured pH during prolonged 10% CO₂ stimulation. Supplementary Fig.5 shows that pH progressively decreased by ~0.1 pH unit during the prolonged stimulation and did not show any sign of recovery. Similarly, breathing frequency increased and reached a sustained plateau without adaptation, a response that mirrors what occurs during briefCO₂ stimulation. Thus, the persistence of NVC during the CO₂ challenge cannot be explained by a pH recovery.

5) *Whether the NVC and CO₂ pathways occlude or not would be better demonstrated by showing they have different mechanisms. If the effects do not occlude, the authors could show that NVC is sensitive to certain blockers that CO₂ is not sensitive too, and that CO₂ is sensitive to certain blockers that do not affect NVC. I am aware of previous literature showing that CO₂-induced vasodilations are*

sensitive to compounds that affect NVC, such as COX inhibitors and NOS inhibitors, but this becomes difficult when considering direct mediator vs modulator effects, especially with NO. Nevertheless, are there blockers that can select for one mechanisms over the other? If so, this would strengthen the authors argument.

Referee#1 is right to stress that most drugs affecting CO₂ responses also affect NVC, as showed in Hosford et al. meta-analysis. Unfortunately, to our knowledge, there is no selective drug for one pathway without affecting the other one, but it is important to remind that the precise mechanisms underlying CVR to CO₂ are not fully understood. As mentioned above, our goal is not to decipher the differential signaling pathways involved in both responses. Our work is purely based on the timing of the initial response of NVC (for briefCO₂) and on the overall shape of NVC (for continuous CO₂), this showing that NVC persists during CO₂/pH acidosis.

Nonetheless, although we could not properly address this point, our data on timing (see the paragraph in discussion below) suggest that the mechanisms underlying NVC and reactivity to CO₂ may be different.

Discussion: “The timing of events observed revealed that relaxation of SMCs lagged by ~3-4s after the drop in GCaMP6 fluorescence (i.e., cell acidosis), in contrast to what occurs during NVC, where relaxation occurs ~400 ms after a true decrease in calcium⁹. The signaling mechanisms linking acidosis to relaxation are thus distinct, slow and beyond the scope of our study. Overall, our data suggest that hypercapnia and thus probably endogenous CO₂ produced by neurons do not affect NVC, which principally depends on transmitter release and a rise in extracellular potassium¹⁰⁻¹². In addition, it shows that NVC and CVR to CO₂ should be considered as two stimuli exploring distinct cerebrovascular reflexes with different dynamics, NVC being extremely rapid and CVR to CO₂ requiring a couple of seconds to occur.”

If the pathways do occlude in some condition, such as at the penetrating arteriole, then it would be fitting to test the mechanism proposed by the Hosford paper. Use an AAV to knockdown the NaHO3 co-transporter in astrocytes and see if this protein is involved in NVC or not or not. I recognize that this is not a trivial experiment to do. Perhaps the authors could try this or the pharmacology tests described above? I am not saying the authors must have manipulations like this, but it would strengthen their arguments and this paper considerably.

As mentioned above, there is no occlusion of NVC by CO₂, neither at the capillary or at the arteriole levels (see our responses to the previous comments).

6) The sedative dexmedetomidine has both blood pressure and respiratory effects. Blood pressure first raises above normal followed by a decrease below normal values. Respiration is also depressed in the drug. Do the authors know what phase their experiments were in: hypotensive vs hypertensive? This caveat should at least be discussed.

Referee #1 is right that all anesthetics have side effects. As we were particularly interested at the idea of measuring "non-invasively" blood pressure, heart rate and respiration during briefCO₂, the first author went to the laboratory of Jessica Filosa (Augusta University) and did experiments with Philip O'Herron and Jessica Filosa in C57/B6 mice, implanted chronically with devices measuring these parameters. These new data (Supplementary Fig.2) show that our experiments were performed when

the mice were lightly hypotensive (90 mmHg, vs 100 mmHg in non-anesthetized awake C57/Bl6 mice^{13,14}), with a bradycardia (260 bpm, vs 550 bpm in awake mice) and with a mild depression of respiration rate (2.4 Hz, Figure 1c and d, vs ~3 Hz in awake mice). Interestingly, these new experiments show that CO₂ causes a transient small drop of blood pressure (1 mmHg) that can explain the early decrease of velocity observed in pial arterioles and in the carotid, followed by a rebound. CO₂ also increases respiration during the entire stimulation. So, as expected CO₂ stimulation affects the peripheral vascular homeostasis, in addition to brain cerebrovascular reactivity. These new data are discussed line 121-123 and 228-229. Philip O'Herron and Jessica Filosa are now authors in the manuscript.

On this note, if the aim here is to test the same hypothesis as the Hosford paper (but with different imaging modalities), the authors here should make sure the anesthetic/sedative is the same. It is expected that every anesthetic or sedative has some impact on cerebral blood flow and its control. Looking it up, that paper used α -chloralose. The authors here should test their main effect in α -chloralose to rule out this caveat.

As mentioned above every anesthetic has an impact on blood flow, and more generally, on brain function. Besides the anesthetics, there are several significant differences between our experimental paradigms: Hosford et al. used rats (except for the last experiment using NBCe1 KO mice), acute preparation, ventilated animals, 60 seconds electrical stimulation of the forepaw. We have proposed, in the Discussion, that among several reasons, one possible explanation for the discrepancies between Hosford et al.'s work and ours is that hypercapnia is known to be an analgesic¹⁵, increasing pain threshold. We hypothesize that the response to the electrical stimulation with subcutaneous bipolar electrodes in the forepaw (300 μ s pulse, 3 Hz, 1.5 mA, 60 s) generates a large pain component in their NVC response, which is decreased by 5% CO₂ and then almost abolished by the 10% CO₂. Indeed, they had to increase the intensity of the stimulation (Fig. 3) and under this condition they regained a response. Reviewer#1 must realize that our goal was not to reproduce Hosford's work but to introduce a new CO₂ stimulation and then test if CO₂ produced by neurons participate to NVC under more physiological conditions, i.e. light sedation, a brain state that is closer to awake than α -chloralose, freely breathing animals, a mild sensory stimulation. Unfortunately, this issue cannot be investigated in awake animals because CO₂, even at concentration as low as 5% during few seconds, is a stressful stimulus that makes the animals run even after weeks of training. Since running has a major impact in brain oxygenation and blood flow^{8,16}, we could not discriminate between vascular response to CO₂ or to locomotion.

Minor Comments:

I need clarification on the z score approach. The following formula is given: $z \text{ score} = (x - \mu_{\text{baseline}}) / \sigma_{\text{baseline}}$. Mu is not defined but I assume it is the mean of the baseline values. Is that correct? The authors report "SD" as their primary measure in most of their graphs. Is it truly standard deviation that the authors are plotting here or should it be better labelled as "z score". Plotting standard deviation over time makes little sense to me.

The reviewer is right: the formula is correct, μ_{baseline} is the mean value of the baseline, and σ_{baseline} is the standard deviation of the baseline. It is now written more precisely in the Methods. The Y axis

represent the Z score. We changed the Figures accordingly and now indicate 1 for one unit z score. Plotting the data in Z score has a single but major advantage: it decreases the role of vasomotion and when studying the onset of any given response, it improves the threshold detection. We found this approach crucial (see¹) to determine precisely the differential onsets of NVC along the vascular arbor in the cortex. In the current paper, it was even more important as we could not average as many CO₂ versus whisker stimulations, for the former responses were prolonged and we did not want to harm the mice. The determination of the onsets and the response amplitudes thus required to express the data in Z score. Note that in the olfactory bulb where vasomotion is less prominent, this approach is less necessary.

It would be nice to see what the Alexa488 signal and H-Ruby signal look like separately and as a ratio.

It is a good idea and Referee #2 similarly asked to see raw data for both the H-Ruby and AF488 signals. We now show a representative data in Supplementary Fig.3. It clearly shows that the dual simultaneous measurements were important to eliminate the bias due to dilation and the absorption of excitation light by hemoglobin. This point is detailed in our response to Reviewer#2 (comment 4) that we copy-paste below:

We totally agree that vessel dilation is a huge problem in fluorescence measurements in vessels. However, we would like to stress that the technical approach used by Fan et al. 2020, (scanning with a Bessel focus with an extended axial FWHM of 67 μm) is responsible of the fact that dilation increases fluorescence. It is clever and allows to monitor multiple vessels in a large field of view. In our case, it is the other way around as scanning a Gaussian focus with an axial FWHM of few μm generates the opposite phenomenon: dilation causes a decrease of fluorescence of inert dyes because of light absorption by hemoglobin (as shown in the new Supplementary Fig 3). When the focus is centered in a large vessel, dilation above the focus absorbs excitation light. This emphasizes the necessity of using a ratiometric approach, with a pH sensitive and another inert dye. We have presented a paper on the issue at the last SPIES meeting in January 2024¹⁷. To conclude, during dilation and for a Bessel focus, the increase of dye in the excitation volume (extended axial PSF) dominates whereas for a small Gaussian focus, the hemoglobin absorption above the excitation volume dominates. We now provide raw data for a typical experiment, with separate H-Ruby and AF488 traces (plus the ratio of the traces) (Supplementary Fig. 3). These data demonstrate the issues of light absorption by hemoglobin.

In fig 4g and fig 5e, where the % change is plotted in constant CO₂, I am confused what is being plotted here exactly. Is this the baseline increase to CO₂? Or is it the evoked NVC response from the new baseline in CO₂? I assume the latter, and if so the connecting line between the points is confusing.

We have now clarified the point in the Legends. What is plotted is the increase of resting RBC velocity or resting CBV to continuous CO₂ stimulation (10%).

Line 76 “which enabled us to uncover”

Corrected

Line 112, “imposes” should probably be “implies”

corrected

Responses to Reviewer #2:

General comments:

This is a very well written and carefully conducted study investigating the role of CO₂ in neurovascular coupling responses. CO₂ has a potent vasodilatory effect, but it is controversial as to whether normal production of CO₂ during brain metabolism is a dilatory signal to brain arterioles. The authors point out that many prior CO₂ inhalation studies used a "sledge hammer" approach with long duration, high level CO₂ (minutes) and anesthetized preparations. In this respect, the authors take an innovative new angle using different animal preparations (chronic windows, sedation) and brief delivery of CO₂ (carefully timed short duration CO₂ at time-scales relevant to neurovascular coupling). The key experiment determines whether brief CO₂ inhalation has any effect on whisker-induced neurovascular coupling responses. In contrast to prior studies, the authors find no evidence that CO₂ inhalation affects neurovascular coupling. Overall, I think this is a very elegant study and I concur with most of the presented findings. However, there are some measurements that remain difficult for me to interpret and could use further clarification/analysis.

We thank Reviewer #2 for his very positive comments.

Major comments

1) The authors show an interesting decrease in average GCaMP6/8 fluorescence across all neurovascular cells and attribute this to pH sensitivity of the calcium sensor and CO₂-mediated acidosis. However, the signal change may reflect some real intracellular calcium change in the cell. For example, with smooth muscle cells, a decreased in calcium is consistent with vasodilation. How do the authors rule out the possibility that GCaMP signal change is dependent upon both calcium change and acidosis in their studies?

We agree that we cannot exclude that part of the fluorescence decreases in all cell types reflect real calcium drops. It is particularly true for SMCs, which in response to sensory stimulations or neuronal activation show a decrease in calcium prior to dilation^{2,9,18}. In the olfactory bulb, the drop of calcium due to NVC starts less than 400 ms before arteriole dilation. This brief delay is thus 7 times smaller than the delay (~3 s) occurring between the fluorescence drop and dilation due to brief CO₂. Still, this does not mean that calcium does not decrease upon CO₂ but if it does, the signaling pathway is very different from the one triggered by neuronal activation. Concerning endothelial cells and astrocytes, calcium decreases have not been reported previously, to our knowledge. This strongly also suggests that the fluorescence decreases were unrelated to calcium, and resulted from a pH decrease.

Our Discussion (lines 242-246) is now stressing our point:

“Acidification of NVU cells was fast. Our measurements are indirect as they are based on decreased fluorescence of GCaMP6/8 in NVU cells. Because this decrease occurs in all cells, and it is known that pH changes, as large as those caused by brief CO₂ (0.15 of pH unit), can lead to a decrease in fluorescence of GFP or Ca²⁺ protein sensors such as GCaMP or R-GECO^{19,20}, we believe that they report intracellular acidosis rather than true calcium changes, although some calcium changes may be masked by acidosis, in particular in SMCs. In these cells, the drop of calcium due to NVC starts less than 400 ms before arteriole dilation⁹. This brief delay is thus 7 times smaller than the delay (~3 s) occurring between the fluorescence drop and dilation due to brief CO₂. This indicates that if some calcium decrease occurs upon CO₂, it involves a signaling pathway different from that triggered during NVC.”

2) *If GCaMP6/8 signal does encode some information about cellular calcium, then the analysis of the fluorescence signals may be too simplistic. Calcium signal patterns in endothelial cells, astrocytes, and neurons are spatiotemporally complex. For example, endothelial cells have a variety of different calcium event types (<https://www.science.org/doi/10.1126/sciadv.abh0101>). The meaning of an average signal decrease is not obvious, especially for cells such as endothelium and astrocytes. Even for neurons, one wonders if decreased neuropil calcium is indicative of decreased pyramidal neuron firing.*

We agree that calcium signals in astrocytes and endothelial cells are cell specific. We participated in the controversy²¹ where the potential role of astrocytes in NVC was discarded. We demonstrated that astrocyte processes, in contrast to astrocyte somata, showed fast calcium responses compatible with a role in NVC, even though there was no technical means at that time to demonstrate a causal role. Following studies using GCaMP targeted to the membrane and uncovering calcium hot spots in astrocytes, as in endothelial cells or SMCs, clearly indicated that before stating that a given stimulus does not modulate calcium signaling, one has to investigate all types of spontaneous calcium signals. What we simply say is that the mean resting fluorescence of GCaMP6 in astrocytes and endothelial cells decreases upon brief CO₂ and it can be explained by a pH effect. Again, this does not rule out that some calcium hot spots are, or are not, modulated during NVC or brief CO₂. This question is far beyond the scope of our study. We now indicate the point in our Discussion (lines 246-247):

“Note that it is also possible that CO₂ affects the frequency of some calcium hot spots in astrocytes, endothelial cells or SMCs, as our approach was not developed to investigate this question.”

3) *Presumably, the level and duration of CO₂ should not cause any swelling of brain tissue. A confirmation of this would be good to have since shifting of the focal plane by change in brain size may affect GCaMP6/8 fluorescence intensity.*

We observed very rare cases of shifts in the z plane during our experiments, which we discarded, as mentioned in the Reporting Summary form. We are aware of the problem of movement to measure dilation or GCaMP fluorescence in cells near the vessel lumen, i.e., mural and endothelial cells. We now provide three Movies representing what occurs during CO₂ stimulation (Supplementary Movie 1, 2 and 3). In the first one, we describe our approach to measure calcium in a penetrating arteriole during dilation. We show that we first stabilize part of the vessel wall with an auto correlation algorithm. We then measure the fluorescence in the stabilized ROI. In the second video, we show a pial arteriole during a prolonged CO₂ stimulation: CO₂ generates a large dilation with a strong shrinkage of the perivascular space (in dark). With such acquisition, it is indeed impossible to ensure that there is no effect of a movement in z, beside that fact that we specifically choose the z position where the vessel had the largest diameter before applying the stimulation. Still to convince you, we provide a last video, from an animal that had been imaged for several experiments. This animal shows some autofluorescence spots outside the vessel. What can be observed is that during dilation, the spots at a distance further than 20-30 μm don't move. Thus, brain swelling does not underly our calcium measurements.

4) *The H-Ruby dextran is an innovative idea. If I understand correctly, H-Ruby fluorescence increases with greater acidosis. However, fluorescence of an inert dye can also increase as vessels dilate, and this is particularly evident with approaches using broad integration of signal in the axial dimension (<https://www.nature.com/articles/s41467-020-19851-1>). Appropriately, the authors use a ratiometric approach with AF488 as a concurrent signal in a second channel. But no raw data is shown on how these two signals behave with respect to each other. Can the authors please show raw data traces? Can they be certain that all the H-Ruby signal change is solely due to pH fluctuation and not luminal volume change?*

We totally agree that vessel dilation is a huge problem in fluorescence measurements in vessels. However, we would like to stress that the technical approach used by Fan et al.²² (scanning with a Bessel focus with an extended axial FWHM of 67 μm) is responsible of the fact that dilation increases fluorescence. It is clever and allows to monitor multiple vessels in a large field of view. In our case, it is the other way around as scanning a Gaussian focus with an axial FWHM of few μm generates the opposite phenomenon: dilation causes a decrease of fluorescence of inert dyes because of light absorption by hemoglobin (as shown in the new Supplementary Fig 3). When the focus is centered in a large vessel, dilation above the focus absorbs excitation light. This emphasizes the necessity of using a ratiometric approach, with a pH sensitive and another inert dye. We have presented a paper on the issue at the last SPIES meeting in January 2024¹⁷. To conclude, during dilation and for a Bessel focus, the increase of dye in the excitation volume (extended axial PSF) dominates whereas for a small Gaussian focus, the hemoglobin absorption above the excitation volume dominates. We now provide raw data for a typical experiment, with separate H-Ruby and AF488 traces (plus the ratio of the traces) (Supplementary Fig. 3). These data demonstrate the issues of light absorption by hemoglobin.

5) *Fig. 4d. There is a delayed second peak in the vasodilatory response of control mice. Is this a consistent finding in the author's stimulation paradigm? If so, there seems to be a deviation between experimental groups at this phase, and a separate analysis of this phase may be warranted.*

We apologize for the confusion this particular experiment raised. The reason of the delayed peak was also raised by reviewer #1. To clarify the issue, we have changed Fig.4, added a supplementary figure (Supplementary Fig. 4) and modified the text in results and discussion. See below:

In Results: “Fig. 4c shows that neuronal responses (perivascular bulk calcium; Thy1-GCaMP6s mice) were not affected during CO₂ diffusion and acidification of the periarteriolar column. Note that the drop of GCaMP6 fluorescence following neuronal activation resulted from CO₂-induced acidosis and not from a delayed inhibition. In addition, the superposition of arteriole dilations in both conditions show that the first phase of dilation (the first 5-6 s leading to the dilation peak) was similarly not affected by CO₂ (Fig. 4d). To analyze the extent to which NVC and CO₂ vascular responses were additive or not, we compared the calculated summation of these responses measured separately (Fig. 4e) with the experimental summation of responses to both stimuli, with whisker stimulation occurring either simultaneously with briefCO₂ (Fig. 4f) or with a 2 s delay (Fig. 4g). Fig. 4f-h shows that the calculated summation of separate responses and the experimental summation with the two stimulation paradigms are similar when considering the first 5-6 s of NVC (up to the response peak). This means that there is perfect additivity and no occlusion of the two responses during this period. Thus, the early period of CO₂ diffusion and acidification of the NVU does not influence the initiation of NVC, although it may influence the second phase of NVC (see Supplementary Figure 4 and Discussion).”

In Discussion (lines 281-289): “However, NVC clearly occurred during acidosis and CO₂ diffusion: NVC and CO₂ responses showed full additivity during the first 6 seconds of NVC. A delayed second phase (or hump) appears during "paired" stimulation experiments. The shape of the hump depends on the timing of the summation. However, the difference between experimental and calculated summation of both NVC and CO₂ responses suggests a possible modulation of this late phase by acidosis (Supplementary Figure 4c). Although the investigation of this possible modulation is beyond the scope of our study, many mechanisms could be hypothesized, such as the degree of actin/myosin filament overlap in SMCs at the onset of second dilation or an effect of CO₂ on the mechanisms underlying the undershoot that terminates the NVC response.”

6) *The authors contrast their findings with the recent study by Hosford et al. It seems to me that a major difference is the magnitude of resting state CBF change induced by CO₂ in the current study and that of Hosford et al. Hosford et al. show a ~2-fold increase in resting state CBF change (ASL-MRI) in their paradigm, and the current study shows ~1.3-fold change in velocity even with prolonged CO₂ (Fig. 4g). If prolonged, high level CO₂ maximally dilates arterioles then there is no longer a reserve for neuronally-evoked dilation. In contrast, the current study uses a more finessed approach that preserves some room for further arteriolar dilation. It is actually hard to make a true head-to-head comparison of how CO₂ in the two studies is changing resting state CBF with the data provided. Can the authors comment on whether this conclusion is valid?*

In Hosford et al., experiments were performed in rats, ventilated under α -chloralose anesthesia. In our case, mice were just sedated, free to hyperventilate, which lowers arteriole PCO₂. These differences can explain why, under continuous 10% CO₂, we observed 39 % increases of RBC velocity in capillaries, 62% increases of flow in pial arterioles and 40 % increases of CBV (with fUS), whereas Hosford et al. reported ~100% increases of blood flow with ASL MRI. One important point is that although our mice were free to hyperventilate during long CO₂ stimulation, resting respiration frequency and pH reached plateaus. Thus, we can estimate that our CO₂/pH stimulation was stable. We take the message from reviewer #2 that our stimulus was maybe more "physiological" but lighter. It is to note that in Hosford et al. the addition of acetazolamide on top of the 10% CO₂ stimulation further increased blood flow up by a factor~3, indicating that even under their condition, maximal dilation was not reached. To conclude, we believe that we used a stimulation protocol which would already be considered as "extreme" in clinics and if RBC velocity and CBV changes evoked by neuronal activation are unaffected by this condition, it is hard to believe that CO₂ is involved in NVC, at least in its initiation. We now discuss more thoroughly the conditions differences in the Discussion.

Discussion: “A major difference between the two studies is that our whisker stimulation is a mild sensory stimulation while paw electrical stimulation likely stimulates pain fibers in addition to sensory fibers, and it is known that hypercapnia elevates pain threshold and depresses nociception¹⁵. Another difference may reside in our chronic sedated mouse preparation versus the α -chloralose anesthetized ventilated rat model. In our preparation, animals were free to hyperventilate to compensate acidosis and intravascular pH dropped by 0.1 pH unit while parenchymal pH decreased by 0.2 unit in Hosford's experimental conditions during prolonged CO₂ stimulation. This could also account for some of the differences. Still, even if our acidosis is lighter, it is still important and NVC is perfectly preserved in the different vascular compartments.”

Minor comments

Fig. 4e. Please provide a more in depth explanation of the "occlusion hypothesis". The schematic may be too simplified for readers to quickly understand this concept.

The schematic has been modified and the caption explains better the tested hypothesis as followed "Schematic of the hypothesis: Top, CO₂ produced by neurons initiates NVC. Bottom, exogenous CO₂ (briefCO₂ or continuous CO₂ stimulation) diffusing from vessels to the parenchyma should lower extracellular pH and modulate the role of endogenous CO₂ on NVC."

The Results section has been modified as followed: "In the somatosensory cortex, long CO₂ stimulation (10%, 10 min) was shown to result in markedly decreased NVC, an effect attributed to the saturation of CO₂-dependent processes (i.e., "occlusion") triggered by neuronal activation and resulting in NVC²³. The timing of briefCO₂ makes it possible to test the hypothesis that exogenous CO₂ would alter NVC as CO₂ diffusion and acidification of the periarteriolar column precedes arteriole dilation by several seconds."

Figure 1,2,3, in the time-course panels, please clarify what the downward pointing arrows mean. I presume it means the point of significant change from baseline, but this is not specified in the legend.

The arrows represent the onset (time to reach 10% of the fit) of the curve (mean data) presented. It is now specified clearly in all legends as followed "The arrow indicates the onset time to reach 10% of the fit of the dilation curve."

Results: In the calcium imaging studies, it is not entirely clear that the varied NVU cell types imaged were experiments in different types of mice. This could be made more clear in the Results section, as readers may think that different NVU cells are being measured concurrently, raising concern of contaminating signals between cells since all the GCaMP6/8 traces look similar.

We agree that it was confusing. We have clarified the point in both the results section and the caption.

Results: Lines 130-131: "To investigate which cell type contributed to the arteriolar dilation induced by briefCO₂, we measured the dynamics of GCaMP6 and GCaMP8 fluorescence in each cells of the arteriole neurovascular unit (NVU), i.e., in endothelial cells, SMCs, astrocyte end-feet and perivascular neurons in different strains of transgenic animals (Fig. 3a,b)

Caption: "**b**, Upon briefCO₂ stimulation, dilation of the penetrating arteriole was preceded by a decrease in GCaMP6 or GCaMP8 fluorescence in all cells of the neurovascular unit (note that fluorescence of each cell type was recorded individually in one of the six transgenic lines expressing either GCaMP6 or GCaMP8), suggesting that it may result from modulation of GCaMP fluorescence due to pH (n = 5-8 vessels, 4-6 mice)."

Responses to Reviewer #3

The main topic of this manuscript is outside my expertise. Based on the editor's request, I hereby provide my evaluation of the new pH probe used in the study.

The authors Tournissac et al. describe in their work the use of a pH responsive rhodamine probe H-Ruby dextran. Design and synthesis of the probe is based on published work (properly cited). The major difference when compared to the previous work is that here the authors used Rhodamine B instead of X rhodamine as the core fluorophore structure. This difference is reflected in slightly different photophysical properties, which however, do not significantly affect the probe's performance. The pH responsive characteristic of the new probe was studied. The probe responds well to pH changes in the range of 10 - 4.5, which is relevant to the study. The synthesis is adequately described in the supporting information. The only missing information in the supplementary information is that the clickable dextran is not labeled in the scheme as compound Nr. 6. Similarly, the click product should be depicted as compound Nr. 7.

Based on the available data, I conclude that the use of the probe is appropriate, even though the design and the synthesis is based on known principles and strategies. All necessary details, such as the synthetic part and pH response evaluation are sufficiently described in the article.

We thank Reviewer #3 for the valuable comments.

As correctly noted by Reviewer #3, the design of the H-Ruby was slightly modified compared to our previous work. This modification was driven by the fact that X rhodamine is flat and hydrophobic which favors the H-aggregation within the dextran conjugate and leads to a shifted apparent pKa values in aqueous media, as we showed in our original paper. To circumvent this issue the X-rhodamine core fluorophore was replaced by a less hydrophobic diethylrhodamine.

Although the present work does not focus on the development of a new pH probe, we believe that this modification helped in accurately measuring pH variation in the blood vessels.

The labeling of the molecules has been completed in the supplementary information accordingly to Reviewer #3's comments.

Responses to Reviewer #4

General comments

The study conducted by Tournissac et al. addresses the fundamental question of CO₂'s role in neurovascular coupling. The authors employed a combination of functional ultrasound and two-photon imaging with pH sensors in the mouse barrel cortex. They proposed an innovative method involving whisker stimulus and hypercapnic challenges to analyze CO₂'s impact.

The findings revealed an initial decrease in blood velocity during a brief CO₂ challenge, followed by a diameter increase a few seconds later. The authors hypothesized that this initial decrease is linked to upstream flow modifications, as indicated by a drop in cerebral blood volume (CBV) observed with functional ultrasound in the internal carotid.

The CO₂ challenge yielded no impact on the response to whisker stimulation leading to the conclusion that CO₂ has no role in the neurovascular coupling.

The article is well-written, with clear research questions and a detailed description of the methodology.

However, I have the following comments that I would like to be addressed:

We thank Reviewer #4 for his very positive comments.

Major comments

1) The relevance of the CO₂ challenge as a model for CO₂ release from local brain activity is unclear. Figure 3 partly addresses this but lacks specific results. Without quantitative units in fluorescence measurements, it's difficult to compare the CO₂ concentrations during brief CO₂ challenges and whisker stimulation (WS). The study lacks an investigation of pH during WS. A quantitative comparison of these curves is essential to justify using the CO₂ challenge as a model for CO₂ release.

Our goal is not to establish the CO₂ challenge as a model to quantify the release of CO₂. We want to use both of our CO₂ protocols to artificially raise vascular and tissular CO₂ (and decrease pH) and investigate if it affects NVC. NVC involves all cells of the NVU and by showing indirectly (and thus only qualitatively) that acidosis of all NVU cells does not affect NVC, we can question the hypothesis that CO₂/pH changes play a key role in NVC. Can we nevertheless draw some conclusion from our semi-quantitative approach, based on GCaMP6 fluorescence? Fig4c show that synaptic activation generates, as expected, a fast increase of GCaMP6 in neurons whereas briefCO₂, a strong CO₂ stimulation, causes a delayed and slower drop of fluorescence (Fig3b, Fig. 4c). Based on these observations, we consider that WS does not significantly decrease intracellular pH in neurons, unless the response is completely masked by the true increase of calcium. Finally, the fact that the continuous 10% CO₂ challenge, does not affect NVC is a strong argument to discard the CO₂ produced by neurons as a major contributor of NVC.

Still we agree with reviewer#4 that it would be great to reinvestigate pH dynamics during a physiological synaptic activation. We are planning this quantitative study of pH during synaptic transmission in vivo, but 1) it will take a couple of years and, 2) it is not in the scope of the current study. Measuring synaptically-evoked CO₂ production and pH changes non-invasively in neurons and in the extracellular space of chronic animals, will require novel imaging tools that report pH and possibly CO₂ in mice chronically implanted with a glass window (or in a thinned skull preparation). We are starting these tool developments and up to now, most pH measurements in rodents have been performed in vitro, or in vivo with an open skull, either with 1) electrodes which report the bulk tissue response (pH of the interstitial fluid) or 2) fluorescent pH sensors that target what occurs locally at the synapse. To conclude, the physiology of brain pH in vivo remains an open field.

2) While the authors investigated CBV in the internal carotid, they did not examine other systemic cardiovascular effects, such as heart rate and blood pressure, which could influence brain flow during the hypercapnic challenge. They also did not discuss how autoregulation might affect flow under these conditions.

Reviewer #4 is right, the mice systemic state during sedation and continuous CO₂ could certainly influence NVC and responses. As Reviewer #1 raised the same issue (comment 6), we copy-paste here our response:

Referee #1 is right that all anesthetics have side effects. As we were particularly interested at the idea of measuring "non-invasively" blood pressure, heart rate and respiration during brief CO₂, the first author went to the laboratory of Jessica Filosa (Augusta University) and did experiments with Philip O'Herron and Jessica Filosa in C57/B6 mice, implanted chronically with devices measuring these parameters. These new data (Supplementary Fig.2) show that CO₂ causes a transient and small drop of blood pressure (1 mmHg) that can explain the early decrease of velocity seen in the pial artery and the carotid (CBV change, fUS), and which is followed by a rebound. So, as expected, CO₂ stimulation affects the peripheral vascular homeostasis, in addition to brain cerebrovascular reactivity. These new data are discussed line 121-123. Philip O'Herron and Jessica Filosa are now authors in the manuscript.

3) The role of ULM images in the study is unclear. The authors quantified blood velocity in the internal carotid, but it's not evident how this data was used in the study. Were these measurements solely for positioning CBV measurements with functional ultrasound?

Reviewer#4 is right. ULM data were solely used to position CBV measurements in the internal carotid. It is now clearly indicated in the legend of Fig 2c:

“A sagittal section of a mouse brain was recorded with ultrasound localization microscopy (ULM) upon microbubbles injection (i.v.). This approach allowed to position CBV measurements in the internal carotid artery (ICA). Note that it also reported resting blood velocity, the color map representing the horizontal component of the RBC speed.”

4) Statistically, a non-significant difference is not equivalent to no difference. How do the authors address this? Additionally, Figure 4h shows some early differences in neuronal response (overshoot) that are not visible when testing area under the curve (AUC) values.

We have now increased the number of experiments to 13 vessels and 6 mice in Fig 4j-n to determine whether neuronal responses were increased or not during continuous CO₂. We also analyzed the responses during the recovery period following CO₂ stimulation to assess whether the potential increase of neuronal activity was real or reflected an evolution with time of the sedated state of mice. As shown in the figure below, of the 3 experiments in the initial Fig4h, in one case the response remained high after CO₂, in another it returned to baseline and, unfortunately, in the last case the recovery was not monitored. Taken together with our additional experiments, we don't think that CO₂ enhances neuronal responses. Note it was previously suggested that it should decrease neuronal activity²³.

Minor comments:

Figure 2, a: Please include diameter measurements in micrometers.

It is now included in Supplementary Figure 1.

Figure 5, a-b: Why is the y-axis scale bar different for high and low velocities? Functional ultrasound cannot detect an increase in slow vascular flows, although two-photon experiments are averaged for n=44 vessels from 25 mice, compared to n=5 mice in functional ultrasound.

The reason was purely esthetical and we put back similar scale bars for all fUS measurements. The extraction of low velocity CBV is noisier and most people don't have the analytic tools to do it. What is remarkable is that for a strong signal at high speed, we see nothing at low speed. Even though fUS is mesoscopic, its resolution is not as sensitive as two-photon imaging to RBC flow in capillaries.

Line 209: The term "fast CBV" is unclear.

We have clarified the point in line 202 of the Results: We first evaluated the contribution of the arteriolar versus capillary compartments by comparing responses of slow (0.5-1.5 mm/s) and fast (> 3.5 mm/s) flowing CBV (i.e., the CBV fraction flowing with a low or high axial velocity⁵) to either whisker stimulation or CO₂ inhalation.

Line 221: The use of commas seems unnecessary.

We removed them.

REFERENCES :

1. Rungta, R. L. *et al.* Diversity of neurovascular coupling dynamics along vascular arbors in layer II/III somatosensory cortex. *Commun Biology* 4, 855 (2021).
2. Institoris, A. *et al.* Astrocytes amplify neurovascular coupling to sustained activation of neocortex in awake mice. *Nat. Commun.* 13, 7872 (2022).
3. Schulz, K. *et al.* Simultaneous BOLD fMRI and fiber-optic calcium recording in rat neocortex. *Nat Methods* 9, 597–602 (2012).
4. Caldwell, H. G., Carr, J. M. J. R., Minhas, J. S., Swenson, E. R. & Ainslie, P. N. Acid–base balance and cerebrovascular regulation. *J Physiology* 599, 5337–5359 (2021).
5. Boido, D. *et al.* Mesoscopic and microscopic imaging of sensory responses in the same animal. *Nat Commun* 10, 1110 (2019).
6. Chow, B. W. *et al.* Caveolae in CNS arterioles mediate neurovascular coupling. *Nature* 579, 106–110 (2020).
7. Vandal, M. *et al.* Dereglulation of brain endothelial CD2AP in Alzheimer’s disease impairs Reelin-mediated neurovascular coupling. *Biorxiv* 2020.12.10.419598 (2020)
doi:10.1101/2020.12.10.419598.
8. Tran, C. H. T., Peringod, G. & Gordon, G. R. Astrocytes Integrate Behavioral State and Vascular Signals during Functional Hyperemia. *Neuron* 100, 1133-1148.e3 (2018).
9. Rungta, R. L., Chaigneau, E., Osmanski, B.-F. & Charpak, S. Vascular Compartmentalization of Functional Hyperemia from the Synapse to the Pia. *Neuron* 99, 362-375.e4 (2018).
10. Longden, T. A. *et al.* Capillary K⁺-sensing initiates retrograde hyperpolarization to increase local cerebral blood flow. *Nat Neurosci* 20, 717–726 (2017).
11. Attwell, D. *et al.* Glial and neuronal control of brain blood flow. *Nature* 468, 232–43 (2010).
12. Iadecola, C. The Neurovascular Unit Coming of Age: A Journey through Neurovascular Coupling in Health and Disease. *Neuron* 96, 17–42 (2017).
13. Kim, S. M., Mizel, D., Qin, Y., Huang, Y. & Schnermann, J. Blood pressure, heart rate and tubuloglomerular feedback in A1AR-deficient mice with different genetic backgrounds. *Acta Physiol.* 213, 259–267 (2015).
14. Zhao, X. *et al.* Arterial Pressure Monitoring in Mice. *Curr. Protoc. Mouse Biol.* 1, 105–122 (2011).
15. Gamble, G. D. & Milne, R. J. Hypercapnia depresses nociception: endogenous opioids implicated. *Brain Res.* 514, 198–205 (1990).

16. Zhang, Q. *et al.* Cerebral oxygenation during locomotion is modulated by respiration. *Nat Commun* 10, 5515 (2019).
17. Chaigneau, E. M., Tournissac, M., Pfister, S., Collot, M. & Charpak, S. Unbiased quantitative ratiometric measurements in multiphoton microscopy with a generalized model. *Multiphoton Microsc. Biomed. Sci.* XXIV 14 (2024) doi:10.1117/12.2692652.
18. Khennouf, L. *et al.* Active role of capillary pericytes during stimulation-induced activity and spreading depolarization. *Brain* 141, 2032–2046 (2018).
19. Molina, R. S. *et al.* Understanding the Fluorescence Change in Red Genetically Encoded Calcium Ion Indicators. *Biophys. J.* 116, 1873–1886 (2019).
20. Helassa, N., Podor, B., Fine, A. & Török, K. Design and mechanistic insight into ultrafast calcium indicators for monitoring intracellular calcium dynamics. *Sci Rep-uk* 6, 38276 (2016).
21. Otsu, Y. *et al.* Calcium dynamics in astrocyte processes during neurovascular coupling. *Nat Neurosci* 18, 210–218 (2015).
22. Fan, J. L. *et al.* High-speed volumetric two-photon fluorescence imaging of neurovascular dynamics. *Nat Commun* 11, 6020 (2020).
23. Hosford, P. S. *et al.* CO₂ signaling mediates neurovascular coupling in the cerebral cortex. *Nat Commun* 13, 2125 (2022).

REVIEWERS' COMMENTS

Reviewer #1 (Remarks to the Author):

The authors have satisfactorily addressed my comments with new data, updated figures and good arguments. With a clearer demonstration that the NVC and CO₂-induced dilations are additive (updated fig4), I think there is less need now for mechanistic work, and some of my other comments are irrelevant now. As the authors indicate, there may be some interaction in the late phase of NVC, which is interesting, but the effect is insignificant at this point and does not need to be investigated here.

I do not need to see the paper again but this statement in the rebuttal, in the paper and in the legend, I do not understand: Line 174-175 "Note that the drop of GCaMP6 fluorescence following neuronal activation resulted from CO₂-induced acidosis and not from a delayed inhibition." Why isn't this drop due to GABAergic inhibition? I did not see the rationale or explanation for the acidosis interpretation. If the authors do not really know the reason for this drop in the GCaMP signal following neuronal activation they can simply drop the statement.

Reviewer #2 (Remarks to the Author):

The authors have thoroughly addressed my comments, and I have no further additions. This is a carefully conducted and timely study that provides balance to our understanding of the role of CO₂ in NVC.

Reviewer #4 (Remarks to the Author):

I appreciate the authors clarifications concerning the quantification of pH, as well as the supplementary experiments conducted on blood pressure, heart rate, and respiration during brief CO₂ exposure, and the additional analyses provided. The authors have effectively addressed the concerns highlighted in the initial review and have furnished extensive clarifications and supplementary data where required. I recommend that this manuscript be accepted for publication.